# Spatio-temporal variability and decadal trends of snowmelt processes on Antarctic sea ice observed by satellite scatterometers

Stefanie Arndt[1], Christian Haas[1]

[1]Alfred-Wegener-Institut Helmholtz-Zentrum für Polar- und Meeresforschung, 27570 Bremerhaven, Germany

*Correspondence to*: Stefanie Arndt (stefanie.arndt@awi.de)

**Abstract.** The timing and intensity of snowmelt processes on sea ice are key drivers determining the seasonal sea-ice energy and mass budgets. In the Arctic, satellite passive microwave and radar observations have revealed a trend towards an earlier snowmelt onset during the last decades, which is an important aspect of Arctic amplification and sea ice decline. Around Antarctica, snowmelt on perennial ice is weak and very different than in the Arctic, with most snow surviving the summer.

Here we compile time series of snowmelt-onset dates on seasonal and perennial Antarctic sea ice from 1992 to 2014/15 using active microwave observations from European Remote Sensing Satellite (ERS-1/2), Quick Scatterometer (QSCAT) and Advanced Scatterometer (ASCAT) radar scatterometers. We define two snowmelt transition stages: A weak backscatter rise indicating the initial warming and destructive metamorphism of the snowpack (pre-melt), followed by a rapid backscatter rise indicating the onset of thaw-freeze cycles (snowmelt).

Results show large interannual variability with an average pre-melt onset date of 29 November and melt onset of 10 December, respectively, on perennial ice, without any significant trends over the study period, consistent with the small trends of Antarctic sea ice extent. There was a latitudinal gradient from early snowmelt onsets in mid-November in the northern Weddell Sea to late (end-December) or even absent snowmelt conditions in the southern Weddell Sea.

We show that QSCAT Ku-band (13.4 GHz signal frequency) derived pre-melt and snowmelt onset dates are earlier by 20 and

18 days, respectively, than ERS and ASCAT C-band (5.6 GHz) derived dates. This offset has been considered when constructing the time series. Snowmelt onset dates from passive microwave observations (37 GHz) are later by 14 and 6 days than those from the scatterometers, respectively.

Based on these characteristic differences between melt onset dates observed by different microwave wavelengths, we developed a conceptual model which illustrates how the seasonal evolution of snow temperature profiles may affect different

microwave bands with different penetration depths. These suggest that future multi-frequency active/passive microwave satellite missions could be used to resolve melt processes throughout the vertical snow column of thick snow on perennial Antarctic sea ice.

# 1 Introduction

Sea-ice extent in the Southern Ocean has experienced large seasonal and inter-annual variations during the last decades, while ice extent has changed little on decadal time scales (Parkinson and Cavalieri, 2012;Stammerjohn et al., 2012;Turner et al., 2015). In addition, there are strong regional differences, with ice extent increases of about 3.9% per decade in the Ross Sea, and decreases of 3.4% in the Bellingshausen and Amundsen Seas (Turner et al., 2015). This is in strong contrast to the Arctic, where sea ice extent has decreased strongly everywhere in all seasons (e.g. Meier et al., 2014). The strong seasonal variability of Antarctic sea ice impacts processes and interactions between atmosphere, sea ice and ocean, and is a key component driving the polar marine ecosystem (Massom et al., 2001). The presence of snow on the ice dramatically alters these interactions through its impact on ice thermodynamics, mass balance, and light transmission.

The role of snow on Antarctic sea ice is exacerbated by the fact that the ice remains snow-covered in most regions throughout the summer (Massom et al., 2001). However, during spring and summer the entire snow column experiences substantial seasonal changes in its physical properties associated with variations in snow temperature profiles, liquid water content, grain size distribution, and stratification (e.g. Haas et al., 2001;Massom et al., 2001;Nicolaus et al., 2009). Detecting these variations in the snowpack on Antarctic sea ice is highly relevant as they modify the energy and mass budgets of sea ice, and strongly influence the retrieval of sea-ice parameters from satellite remote sensing algorithms (e.g. Willmes et al., 2014).

In contrast to sea ice in the Southern Ocean, snow and sea ice properties in the Arctic undergo strong changes during the seasonal cycle (Sturm and Massom, 2017). Due to different radiation and turbulent flux regimes of the Arctic atmosphere during the spring-summer transition (e.g. Andreas and Ackley, 1982;Nicolaus et al., 2006) liquid water forms usually rapidly within in the snow. The subsequent positive albedo-feedback processes accelerate seasonal snowmelt and the disappearance of snow, and lead to widespread formation of surface melt ponds resulting in substantial alterations of dielectric properties of the ice and snow surface associated with increasing microwave emissivity and decreasing radar backscatter. This distinct seasonal cycle of surface properties and related microwave signatures of Arctic sea ice has been utilized to identify different snow and sea-ice melt stages and the time of melt onset from passive [e.g. *Markus et al.*, 2009] and active satellite microwave remote sensing observations (e.g. Markus et al., 2009). Those satellite retrievals of melt onset in the Arctic have shown that melt onset occurred earlier by 1 to more than 10 days decade$^{-1}$ since 1979 when routine satellite observation commenced, consistent with warmer air temperatures and accelerated ice retreat in the Arctic during the recent past decades.

However, on Antarctic sea ice, the retrieval of snowmelt onset is more challenging because thawing and melting are weaker and more sporadic than in the Arctic. There is widespread occurrence of diurnal thaw-freeze cycles (Haas et al., 2001;Nicolaus et al., 2006;Nicolaus et al., 2009), or the snow may only thaw during the passage of warm marine cyclones, with the snow refreezing shortly after (Willmes et al., 2006). These thaw-refreeze events cause strong, destructive snow metamorphism with icy snow and ice layers. Under more intensive melting conditions, snow changes from the pendular to the funicular regime (e.g. Denoth, 1980) where the liquid snow melt water percolates through the snowpack to lower, colder layers or to the ice

surface where it eventually refreezes to form superimposed ice (Tison et al., 2008;Haas et al., 2008;Haas et al., 2001;Nicolaus et al., 2009;Willmes et al., 2009).

These different predominant snow processes and the absence of melt ponding result in different sea ice microwave signatures in the Antarctic than in the Arctic. The absence of longer periods with large amounts of liquid water within the snow mean

that strong, persistent decreases of backscatter or increases of emissivity typical for Arctic sea ice during the melt season do not frequently occur in the Antarctic. In contrast, strong snow metamorphism leads to large-grained, polygonal granular textured and salt-free (snow) grains and melt clusters (Colbeck, 1997), increasing radar volume- and surface scattering (Colbeck, 1997;Abdalati and Steffen, 1995;Onstott and Shuchman, 2004), and decreasing microwave emissivity (Willmes et al., 2006).

The presence of frequent diurnal thaw-freeze cycles has been utilized by Willmes et al. (2009) and Arndt et al. (2016) to develop algorithms to observe melt processes and to retrieve snowmelt onset dates from passive microwave radiometers, such as the Special Sensor Microwave/Imager (SSM/I). Their algorithms are based on analyses of the magnitude of diurnal 37 GHz, vertical polarized Brightness Temperature changes observed from ascending and descending morning and afternoon satellite passes approximately 12 hours apart. Both studies found large interannual snowmelt onset variability but no pronounced trends.

So far, there have only been few studies of Antarctic sea ice melt onset using active microwave sensors, i.e. radars. Drinkwater and Liu (2000) applied an Arctic-like algorithm searching for rapid drops in radar backscatter. However, such drops are only occasionally observed on seasonal sea ice near the ice edge, or on the Larsen Ice Shelves (Bevan et al., 2018). On sea ice, drops in radar backscatter can also result from flooding events (Lytle and Ackley, 2001), potentially close to the time of complete deterioration of the ice. Drinkwater and Liu (2000) stated that such backscatter drop events are sparse and short lived.

Therefore, their algorithm may not be easily applicable to most regions of Antarctic sea ice.

In contrast, Haas (2001) used C-band ERS scatterometer data from 1992 to 1999 to show that increasing radar backscatter from winter to summer is typical of Antarctic perennial ice, and is consistent with the theoretical considerations of backscatter from metamorphic snow and superimposed ice discussed above. On average, backscatter increased by 5.6 dB during 96 days, commencing on 15 November.

In this study, we update and extend the work by Haas (2001) with new satellite scatterometer data to study if any long-term changes in Antarctic melt onset on sea ice emerged since 1992-99. For this purpose, we compile time series of European Remote Sensing (ERS)-1/2, QuikSCAT (QSCAT), and Advanced Scatterometer (ASCAT) scatterometer data from 1992 to 2015. A similar time series has been compiled and analyzed with QSCAT and ASCAT data for the Arctic (Mortin et al., 2014).

We apply the melt-onset algorithm of Haas (2001) to both perennial and seasonal ice regimes (Figure 1), and revise it to include early melt season effects on backscatter, defined as pre-melt phase. In addition, we use twice daily QSCAT observations between 1999 and 2009 to retrieve the magnitude of diurnal backscatter changes similar to earlier work with passive microwave observations discussed above (Arndt et al. (2016);Willmes et al. (2009); Figures 2 and 3).

Three are two short overlap periods of ERS-2 and QSCAT in 1999/2000 and of QSCAT and ASCAT in 2008/9. During those we note clear differences in backscatter behavior and melt onset timing observed by QSCAT and the other scatterometers, which we attribute to the different radar frequencies employed by ERS and ASCAT (C-band; 5.6 GHz), and QSCAT respectively (Ku-band; 13.4 GHz), and their different penetration depths (e.g. Ulaby et al., 1986). The time differences between

melt onset detected by the C-band and Ku-band sensors were adjusted to construct the long time series from 1992 to 2015. Finally, we compare results with previously published melt onset dates retrieved from satellite passive microwave sensors using frequencies of 37 GHz (Arndt et al., 2016). The time differences between those retrievals and the radar results are again discussed in the context of temporal snow evolution, penetration depth, and the sensitivity of 37 GHz signals primarily to the uppermost snow layers. The results obtained here demonstrate the potential to observe snow processes at different depths from

space, opening new avenues for multi-sensor studies of energy and mass budgets of snow on sea ice in the Southern Ocean. We would like to point out that with perennial ice we describe sea ice that survives at least one summer melt season. It is important to note that this includes ice that is initially first-year ice, but experiences the melt season to become perennial ice. Antarctic first-year ice is known to be relatively thin and can possess negative freeboard due to its relatively thick snow cover (Massom et al., 2001). Typically, negative freeboard leads to flooding and slush at the snow/ice interface which can refreeze

to form snow-ice (e.g. Eicken et al., 1994; Jeffries et al., 1997; Haas et al., 2001; Nicolaus et al., 2009). Due to the presence of salty slush even cold snow on first year ice can be quite saline and damp in winter, which contributes to its low radar backscatter in the end of winter (Lytle and Ackley, 1996). However, it is important to note that brine in snow percolates downwards as soon as the snow warms, and that during summer snow salinity on perennial Antarctic sea ice is normally low and therefore negligible for satellite microwave observations (Massom et al., 2001; Haas et al.; 2001; Nicolaus et al.; 2009).

In addition, snow ice and superimposed ice formation lead to zero or positive freeboard, and therefore flooding and saline snow are uncommon on perennial ice in summer (e.g. Massom et al., 2001; Haas et al., 2001; Nicolaus et al., 2009), supporting the increases of backscatter during melt onset utilized here.

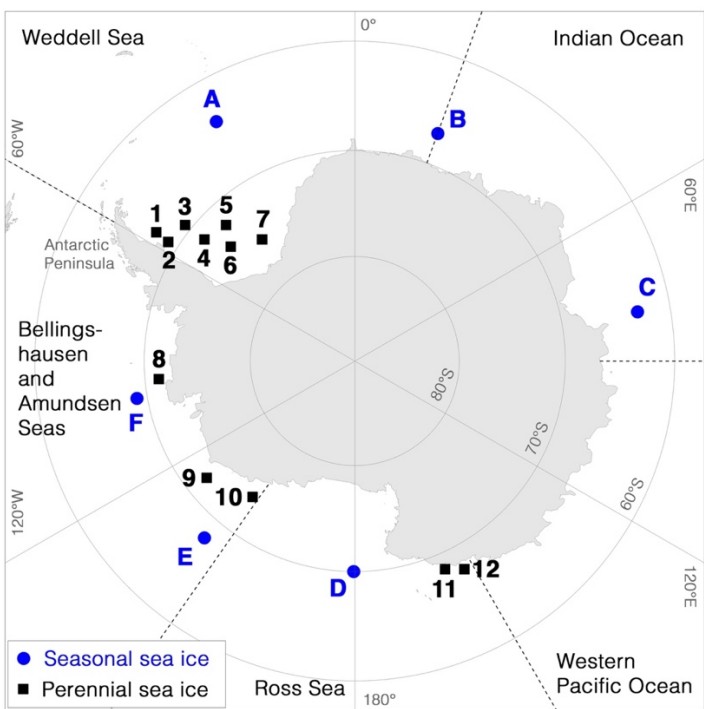

**Figure 1.** Map of Antarctica showing the 12 study locations in the perennial sea-ice zone (black/squares, locations 1-12, same as used by Haas (2001)) and 6 study locations on seasonal sea ice (blue/circles, regions A-F).

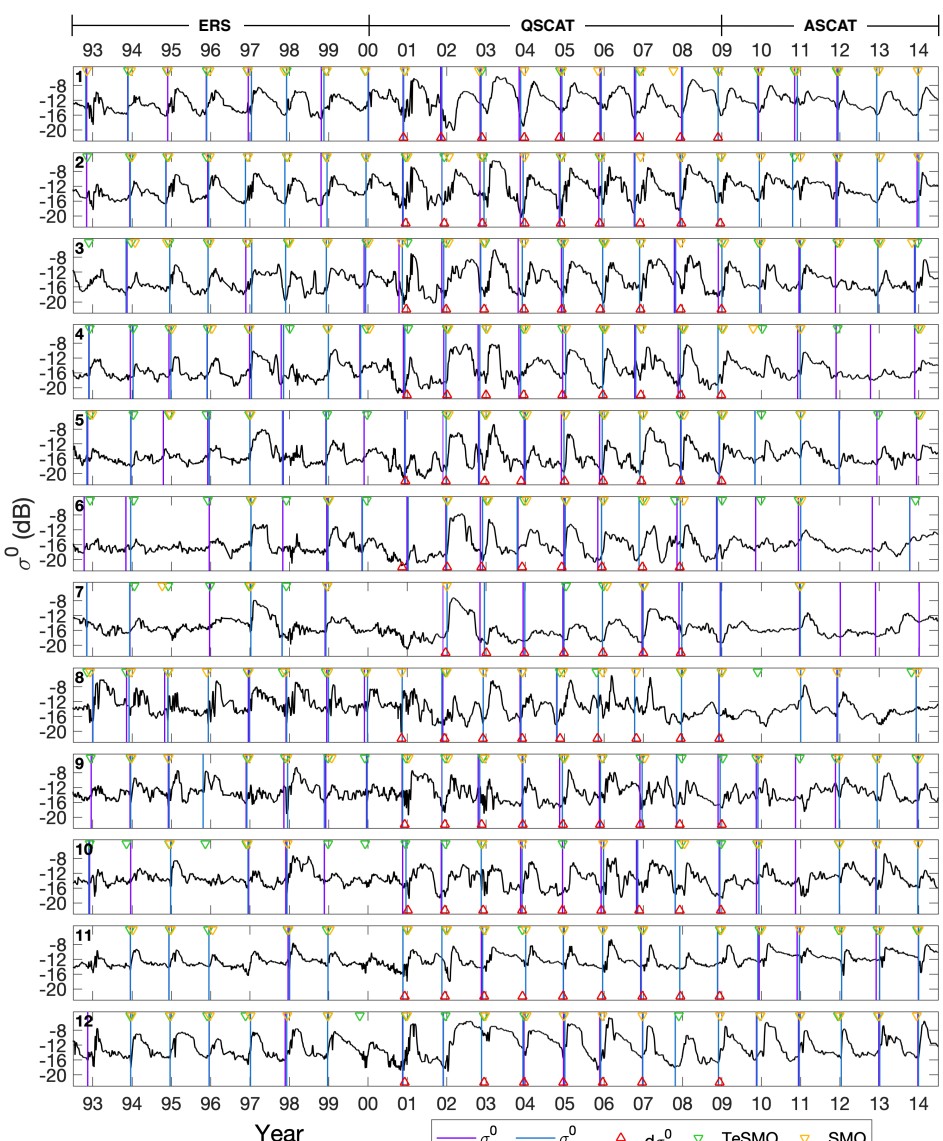

**Figure 2.** Time series of 6-day mean backscatter coefficients ($\sigma^0$) from 1992/1993 to 2014/2015 averaged over the respective 9 pixels at the study locations on perennial sea ice (Figure 1, 1-12). Purple and blue lines show the scatterometer-derived snow pre-melt ($\sigma^0_{pre}$) and melt ($\sigma^0_{melt}$) onset dates when the algorithm criteria were met (Section 2.2.1). Red triangles indicate the snowmelt onset retrieved from diurnal backscatter variations ($d\sigma^0$) of QSCAT data. Green and yellow triangles show the Temporary Snowmelt Onset (TeSMO) and Continuous Snowmelt Onset (SMO) derived from passive microwave observations from Arndt et al. (2016).

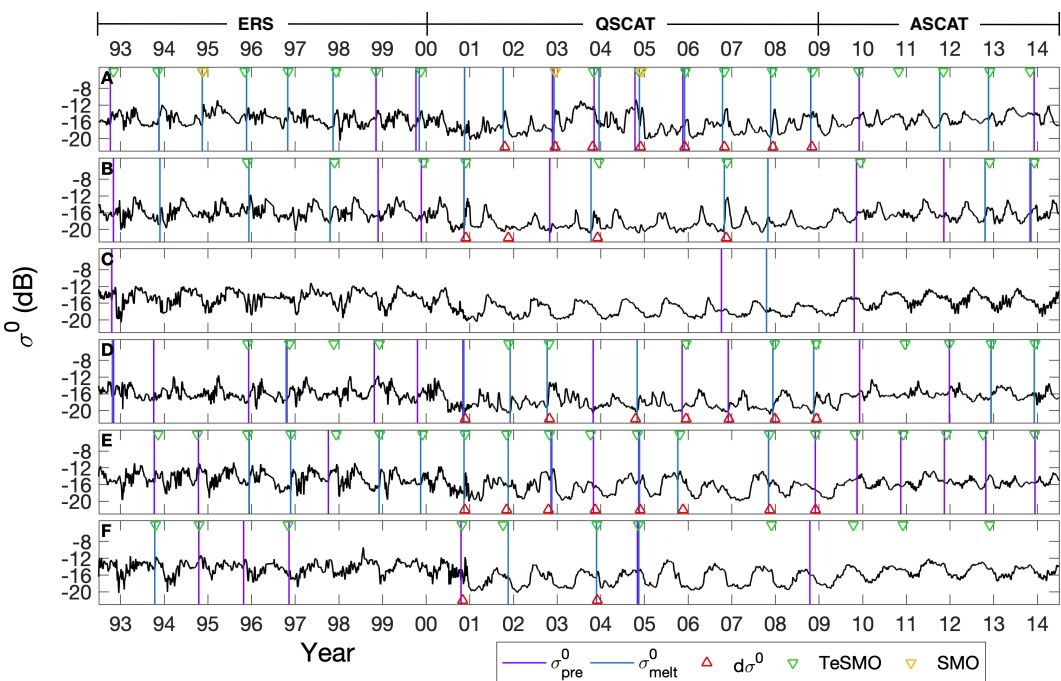

**Figure 3.** As Figure 2, but for study sites on seasonal sea ice (Figure 1, A-F).

## 2. Snowmelt retrieval from satellite scatterometer data

### 2.1 Data set

5    All presented scatterometer data sets were obtained from the NASA Scatterometer Climate Record Pathfinder (SCP) project, sponsored by NASA (http://www.scp.byu.edu/). For the following analysis, all data are interpolated to a 25x25 km$^2$ SSM/I polar stereographic grid, using nearest-neighbor resampling.

The scatterometer data from the European Space Agency's (ESA) European Remote Sensing (ERS) 1 and 2 missions are
10    provided as 6-day averages of vertical co-polarized C-band (5.3 GHz, wavelength of 5.7 cm) backscatter. This low temporal resolution was chosen in order to cover the polar oceans and maintain a reasonable stability of successive backscatter maps from this early mission (Ezraty and Cavanié, 1999). Backscatter $\sigma^0$ is obtained over a 500 km wide swath and normalized to an incident angle of 40°. ERS-1 was operational from 1991 to 1996, while ERS-2 continued observations until 2001.
The NASA QuikSCAT (QSCAT) mission acquired measurements from 1999 to 2009, i.e. overlapping with both the ERS and
15    ASCAT observations by 2 years, respectively (see below). In contrast to the ERS scatterometer, QSCAT operated at Ku-band, i.e. 13.4 GHz, equivalent to shorter wavelengths (2.2 cm). In this study, we used the QSCAT Scatterometer Image Reconstruction (SIR; Long et al. (1993)) backscatter product in both polarizations (vertically and horizontally co-polarized) normalized to a 40° incident angle. The SIR algorithm enhances the spatial resolution of the daily data product to 4.45 km.

Data are available twice daily, with a morning pass between 04:00 and 12:00 and afternoon pass between 12:00 and 20:00 local time. This allows observations of diurnal backscatter variations caused by colder conditions in the morning and warmer conditions in the afternoon.

Finally, the ESA Advanced Scatterometer (ASCAT) was launched in 2006 on the MetOp-A and MetOp-B satellites. The scatterometer is an upgraded successor of the scatterometers onboard the ERS-1/2 platforms mentioned above, operating in C-band at 5.6 GHz in vertical co-polarization (VV). In this study, we also used the SIR product, similar to the QSCAT product, with a spatial resolution of 4.45 km. Due to a lower spatial coverage of ASCAT, the data are provided as 2-day averages.

To avoid biases of sea ice and snow signatures due to signals from larger amounts of open water in each pixel (e.g. Drinkwater and Liu, 2000), sea-ice concentration (SIC) data from Nimbus-7 SMMR and DMSP SSM/I-SSMIS Passive Microwave Data derived with the bootstrap algorithm were used (Comiso, 2000). The data are available daily since 1978 on a 25 km SSM/I polar stereographic grid. During our study period, starting in 1992, ice concentration data of three different SSM/I sensors were used (F11 from January 1992 to 1995, F13 from May 1995 to December 2008, and F17 from December 2006 onwards).

## 2.2 Methods

We restrict our analysis of melt onset dates to 18 individual small study regions of 3x3 pixels of scatterometer data carefully chosen to represent typical ice conditions in seasonal and perennial ice regimes (Figure 1, Haas (2001)). This approach was chosen over more regional analyses to avoid blurring of backscatter signatures due to the large spatial and temporal variability of sea ice properties and drift. To account for small scale variability and ensure stable results, we computed melt onset dates for each pixel and then averaged the melt onset dates. Backscatter time series of individual, neighboring pixels were highly correlated and the results of our algorithm varied by only a few days from pixel to pixel. Like the studies of Markus et al. (2009), Mortin et al. (2014), Willmes et al. (2009), and Arndt et al. (2016) our approach is also Eulerian, i.e. not taking into account ice drift and advection for which there is little accurate data. However, as most Antarctic sea ice is first-year ice at the beginning of the melt season, Eulerian and Lagrangian approaches show similar temporal behavior of melt signals (Fig. 3 in Haas (2001)).

We applied two melt detection algorithms based on two different scatterometer observables: First we analyze daily mean backscatter $\sigma^0$ which is available from all satellites Second, we analyze the magnitude of diurnal backscatter variations $d\sigma^0$, defined as the absolute difference between morning and evening satellite overpasses only available from QSCAT (Section 2.1).

Analyses for both algorithms were only carried out for locations of perennial ice where sea-ice concentration remained above 70% for at least three weeks into the melting season. This avoided contamination of results by wind-roughened water (Drinkwater and Liu, 2000), and effectively eliminated regions of deteriorating, thin ice where surface flooding and break up into small floes and brash ice may occur, e.g. in the marginal ice zone, with competing effects on backscatter evolution.

### 2.2.1 Melt onset retrieval from time series of daily mean backscatter

The seasonal cycle of surface properties of Antarctic perennial sea ice is evident in its backscatter time series with higher backscatter in summer than in winter (Haas, 2001). These are well visible in Figure 2 which show the complete time series compiled for 12 perennial sea-ice regions: While location 1-7 are chosen from north to south in the Weddell Sea, all other locations are distributed throughout the perennial sea-ice regime around Antarctica (Figure 1). As expected before, all perennial sea-ice regions show a distinct seasonal cycle with a sharp increase during spring and a subsequent slow backscatter decrease towards autumn/winter (Figure 4 a). In contrast, the backscatter time series at 6 locations with seasonal sea-ice distributed around Antarctica (Figure 1) are much less systematic and show a much weaker seasonal cycle than those of perennial sea ice (Figures 3 and 4 b). However, Drinkwater and Liu (2000) defined the Antarctic-wide snowmelt onset by a sudden drop in the radar backscatter signal during spring. Their algorithm indicated widely missing onset dates on perennial sea ice, especially in the southwestern Weddell Sea, since radar backscatter behaves differently there than expected by their algorithm.

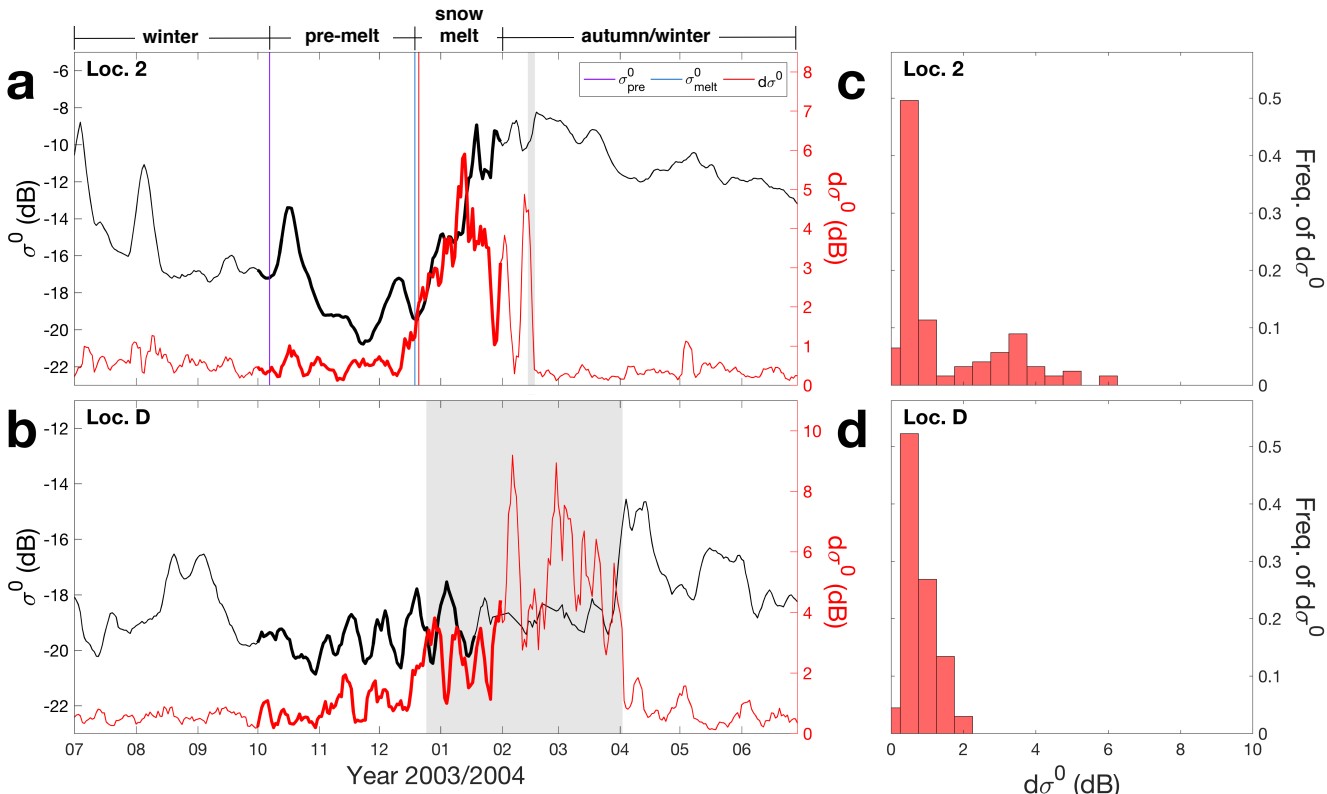

**Figure 4.** Typical annual time series of mean daily QSCAT radar backscatter ($\sigma^0$, black) and its diurnal variations ($d\sigma^0$, red) for location 2 on perennial (a) and location D on seasonal sea ice (b) in 2003/2004. Colored vertical lines illustrate snow pre-melt ($\sigma^0_{pre}$, magenta), and melt ($\sigma^0_{melt}$, blue) onset derived from backscatter time series only, and snowmelt onset derived from

QSCAT diurnal backscatter variations ($d\sigma^0$, red). Grey shaded areas indicate the time period with sea-ice concentration less than 70%, when the algorithm developed here cannot be applied. Bold lines denote the time period for the spring/summer transition retrieval analysis (01 October to 31 January). Histograms in c and d show frequency of occurrence of the magnitude of diurnal backscatter variations $d\sigma^0$.

In the following analysis of the seasonal cycles in the backscatter time series, we distinguish between two major snowmelt stages. Firstly, sporadic, temporal backscatter rises are defined as pre-melt, preceding the larger backscatter rises at melt onset. We interpret these subtle radar backscatter rises to be caused by changing physical properties associated with the spring warming of upper ice layers and with increases in brine volume (e.g. Nandan et al., 2017), or the appearance of traces of liquid

water in the pendular regime in the snowpack. Secondly, the most distinct, largest rise in backscatter during spring or summer is defined as melt onset. At this time, the high intensity of thawing of the snowpack transitions the snow to the funicular regime when liquid water percolates to cause strong snow metamorphism and superimposed ice formation, as observed during various field studies (Haas et al., 2001;Willmes et al., 2009;Nicolaus et al., 2009).. Thaw-refreeze cycles may occur diurnally or with periods of several days. However, on Antarctic sea ice their intensity is usually insufficient to initiate the positive albedo

feedback with accelerated, irreversible snow melt and melt pond formation (Andreas and Ackley, 1982;Haas et al., 2001;Nicolaus et al., 2009). Thawing can also be interrupted by periods of cooler conditions for periods of hours, days or weeks.

In order to retrieve every summer's pre-melt and melt onset, we split the time series into annual sections extending from the

beginning of July of one year to end of June in the following year, with the spring and summer season approximately centered (Figure 4 a, b). Here, the time prior to pre-melt onset is referred to as winter, between pre-melt and snowmelt onset as pre-melt stage, after snowmelt onset as snowmelt stage, and after 31 January as autumn/winter again, as noted in Figure 4 a. Following the approach by Haas (2001), and to be consistent with their analysis of ERS data which are only available as 6-day averages (approximately one week), we first down-sampled the QSCAT and ASCAT data by averaging over 6-day intervals. Then, we

analyze local backscatter maxima and their preceding local minima from three-point running means of each $\sigma^0$-time series. We define the first instance after 1 October when the difference between a local maximum and the preceding minimum is larger than 2 dB as pre-melt onset ($\sigma^0_{pre}$, Figure 4 a, purple line). From there, we search for the instance when the difference between a local maximum and the preceding local minimum is larger than 3 dB. We define that instance as snowmelt onset ($\sigma^0_{melt}$, Figure 4 a, blue line). Note that the algorithm only responds to substantial, long-lived changes of snow microwave

properties due to the 3-point smoothing of the 6-day averaged data. The thresholds are based on the average rise of the backscatter values during the spring/summer transition.

We apply this pre-melt and melt onset detection algorithm to both, perennial and seasonal sea ice. If no backscatter rises of the required magnitude are found during summer, no pre-melt or melt dates are assigned. This may occur in cooler years, on

perennial ice located far south where melt can be small or absent, on seasonal ice when snowmelt is superimposed on other processes like flooding, or where the ice completely disintegrates shortly after snow melt commenced. For example, no melt onset was found at location D in 2003/2004 as shown in Figure 4 b. Overall, pre-melt and melt onset dates were retrievable for 79% and 64% of all analyzed pixels. For the seasonal sea-ice regime, pre-melt and melt onsets were obtained for 46% and

26% of the analyzed pixels.

## 2.2.2 Melt onset retrieval from diurnal backscatter variations observed by QSCAT

Willmes et al. (2009) described the onset of snow surface melt on Antarctic sea ice as the appearance of dominant diurnal thaw-freeze cycles in the surface layer. Analyzing the time series of the absolute difference between two daily brightness temperature values from ascending and descending satellite passes (dT$_B$, 37GHz, vertically polarized), they found an increase

in dT$_B$ once temporary thawing commences in summer.

Since QSCAT data are also available twice a day from ascending and descending passes, the backscatter time series can be utilized to derive diurnal variations at the chosen locations. Therefore, we use the approach by Arndt et al. (2016) to identify the snowmelt onset during austral spring: After applying a 5-day running mean to each d$\sigma^0$ time series between 1 October and

31 January, a histogram of d$\sigma^0$ values with a bin width of 0.5 dB is computed for each location. All histograms only contain data from 1 October until 31 January or until the sea-ice concentration drops below 70% (see Section 2.2). For example, Figures 4 c and d show typical histograms for the respective perennial and seasonal ice study locations. Generally, we found two kinds of histograms: unimodal and bimodal d$\sigma^0$ distributions. A mode is defined as a local maximum bounded by at least one lower bin on each side. Multimodal distributions of d$\sigma^0$ (Figure 4 c) indicate the presence of distinctly different diurnal

backscatter values, which we interpret as different melt stages including differences in the strength of diurnal thaw-freeze cycles. Normally, at least two modes of the d$\sigma^0$ distributions can be observed in those cases. A histogram is considered to be unimodal if a single mode exceeds a fraction of more than 90% of all included data (Figure 4 d). Locations with unimodal distributions are not further considered in the following analysis, as they do not reveal the characteristic diurnal snow backscatter variations. For example, only few occurrences of strong diurnal backscatter variations were found at the seasonal

ice locations, and therefore in most years a reliable melt onset from diurnal variations could not be observed there (Figure 4 b). In contrast, bimodal d$\sigma^0$-distributions are widely detected on perennial sea ice (Figure 4 c). In a next step, an iterative threshold selection algorithm (Ridler and Calvard, 1978) is applied to the respective d$\sigma^0$ time series of each analyzed pixel with a multimodal distribution to derive individual d$\sigma^0$-thresholds delineating winter from summer conditions. Finally, snowmelt onset is defined as the first time when d$\sigma^0$ exceeds the respective local threshold for at least 3 consecutive days

(Figure 4 a).

## 3. Results

### 3.1 Perennial sea ice

**Table 1:** Mean snowmelt onset dates indicated in Figure 2 for the 12 study locations on perennial sea ice shown in Figure 1 (mean ± 1 standard deviation).

| | From scatterometer data | | | From passive microwave data | |
|---|---|---|---|---|---|
| **Region** | **Pre-melt Onset** | **Snowmelt Onset** | **Diurnal thawing-refreezing Onset (2000/01-2008/09 only)** | **Temporary Snowmelt Onset (TeSMO)** | **Continuous Snowmelt Onset (SMO)** |
| Northwestern Weddell Sea (region 1-4) | 23 November ± 21 days | 04 December ± 21 days | 09 December ± 15 days | 12 December ± 16 days | 18 December ± 23 days |
| Southeastern Weddell Sea (region 5-7) | 28 November ± 24 days | 09 December ± 25 days | 16 December ± 14 days | 22 December ± 15 days | 05 January ± 19 days |
| Bellingshausen Sea (region 8) | 28 November ± 23 days | 05 December ± 23 days | 23 November ± 18 days | 28 November ± 20 days | 12 December ± 18 days |
| Amundsen Sea (region 9-10) | 26 November ± 17 days | 05 December ± 19 days | 09 December ± 12 days | 11 December ± 14 days | 23 December ± 16 days |
| Ross Sea (region 11-12) | 10 December ± 18 days | 14 December ± 18 days | 14 December ± 6 days | 14 December ± 13 days | 29 December ± 9 days |
| All regions | 28 November ± 22 days | 07 December ± 21 days | 10 December ± 15 days | 13 December ± 16 days | 23 December ± 20 days |

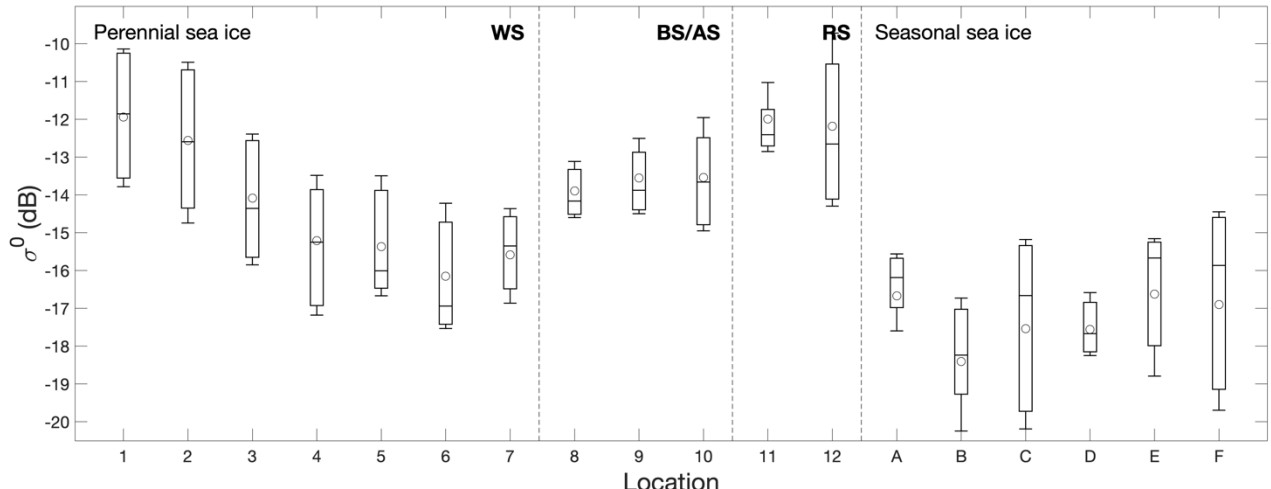

**Figure 5.** Summary of mean radar backscatter σ⁰ (Ku- and C-band) and range of seasonal variability for all study locations during the complete 23-year study period (cf. Figures 2 and 3). Boxes are the first and third quartiles. Whiskers display the 20- and 80-percentiles. Circles indicate mean, horizontal lines median values. Abbreviations according to Figure 1: WS: Weddell Sea, BS/AS: Bellingshausen and Amundsen Seas, RS: Ross Sea.

As expected (Section 1), all backscatter time series show strongly increasing backscatter during the summer, although with some interannual variability (Figure 2). Mean backscatter and the range of seasonal variations are summarized in Figure 5. Mean perennial ice backscatter ranges between -12.0 and -16.2 dB, i.e. is variable between regions and consistently higher than on seasonal ice. Superimposed on mean backscatter are seasonal variations ranging between 1.5 dB at location 8 in the Bellingshausen Sea up to 4.4 dB in the northwestern Weddell Sea and western Ross Sea (locations 1-3 and 12; range between 20 and 80 percentiles in Figure 5). Separating for the different sensor classes indicates that, averaging all 12 study locations on perennial sea ice, the mean amplitude between seasonal minimum and maximum is 8.03dB for the QSCAT time period, while it is only 5.80 and 5.10dB for ERS and ASCAT, respectively (Figure 2), in good agreement with Haas (2001) who studied ERS signals until 1999.

Figure 2 also includes the pre-melt and melt onset dates between 1992/93 and 2014/15 retrieved with the algorithms described. Average pre-melt and melt onset dates in the different regions are summarized in Table 1. Note that in some years melt events could not be observed because the temporal behavior of backscatter changes did not possess the characteristic rises expected by our algorithms. On average, the initial pre-melt onset is on 28 November, 9 days prior to the actual melt onset on 07 December retrieved from rapidly rising daily backscatter and its diurnal variations. In the Weddell Sea, there are strong latitudinal differences with earlier snowmelt onset in the northwest and increasingly later snowmelt onset in the southeast: On average, the earliest pre-melt onsets are observed in mid-November in the northwestern Weddell Sea (locations 2 und 6), and in the Amundsen Sea (location 9), while the latest pre-melt onsets are found in mid-December in the Ross (locations 11 and

12) and southeastern Weddell Seas (location 7). Regions in the northwestern Weddell Sea (locations 2 and 3) and Bellingshausen and Amundsen Seas (locations 8 and 9) had the earliest actual snowmelt dates, at the end of November, while again the southeasternmost region in the Weddell Sea (location 7) had the latest onset dates, in the end of December. Regions in the southeastern Weddell Sea (locations 5-7) and western Amundsen Sea (location 10) frequently do not reveal any distinct pre-melt or melt phases (Figure 2).

Snowmelt onset dates based on diurnal variations in the radar backscatter can only be derived with QSCAT data from 2000/01 to 2008/09. Here, on average, snowmelt onset occurred on 10 December, in an interval from 27 November (northwestern Weddell Sea, location 1) to 21 December (central/northwestern Weddell Sea and Ross Sea, locations 4, 5 and 12).

For comparison, Figure 2 and Table 1 also include Temporary SnowMelt Onset (TeSMO) and subsequent continuous SnowMelt Onset (SMO) retrieved from 37 GHz passive microwave brightness temperature observations [Arndt et al., 2016]. On average, TeSMO occurs on 13 December, i.e. 3 days later than snow melt observed by scatterometers. The average date of SMO occurs another 13 days later on 23 December. On average, the earliest TeSMO starts in mid-November in the Bellingshausen Sea (location 8), while the latest temporal snowmelt onsets are observed in the end of December in the southeastern Weddell Sea (location 6). Again, regions in the northwestern Weddell Sea (locations 1 and 2) showed the earliest SMO in the beginning of December, while the southernmost locations 5 and 6 show the latest SMO in mid-January.

Both, scatterometer and passive microwave observed melt onset parameters consistently show strong gradients towards later snowmelt from north to south as well as the decreasing occurrence of melt events towards the south, in particular in the Weddell Sea. However, we note strong differences in the actual timing of melt events observed by the different sensors which are further discussed in Section 4.4.

## 3.2 Seasonal sea ice

In addition to the perennial sea ice locations, 6 study locations on seasonal sea ice were chosen, distributed around the Antarctic continent (Figures 1 and 3). Mean backscatter ranges between -16.6 and -18.4 dB, i.e. much lower than on perennial ice (Figure 5). In agreement with previous studies discussed in the introduction, none of the seasonal sea-ice regions show any consistent seasonal backscatter cycle. While the Weddell and Ross Seas (locations A and D) show slight increases in backscatter during the transition from winter to summer, backscatter decreases at locations B and C in the Indian Ocean. Locations E and F in the Bellingshausen and Amundsen Seas reveal very little backscatter variation throughout the spring and summer time. However, most regions show a sharp decrease in the backscatter signal with increasing sea-ice melt, consistent with (Drinkwater and Liu, 2000). As expected from the strongly different seasonal backscatter changes over seasonal sea ice compared to perennial ice, snowmelt onset on seasonal ice is only sporadically detected by our algorithm (Figure 3), demonstrating the spurious nature of backscatter signals on seasonal ice. At location A in the Weddell Sea pre-melt is frequently observed (80% of the respective pixels), with an average date of 4 November. In contrast, hardly any pre-melt has been observed at locations C (7%) and F (13%) in the Indian and Bellingshausen Seas. Overall, subsequent actual melt onset is rarely observed either, but most frequently occurring in the Weddell Sea. During the 23-year study period it was observed in 49% of the analyzed pixels with

an average melt onset date of 11 November. Snowmelt onset over seasonal ice could hardly be derived from passive microwave observations either (43%, Arndt et al. (2016)). However, the years and regions when and where it was possible were similar to the ones when also the scatterometer data yielded results. On average, melt onset was observed on 18 November with the passive microwave data (Figure 3).

## 3.3 Compilation of snowmelt onset time series and analysis of long-term trends

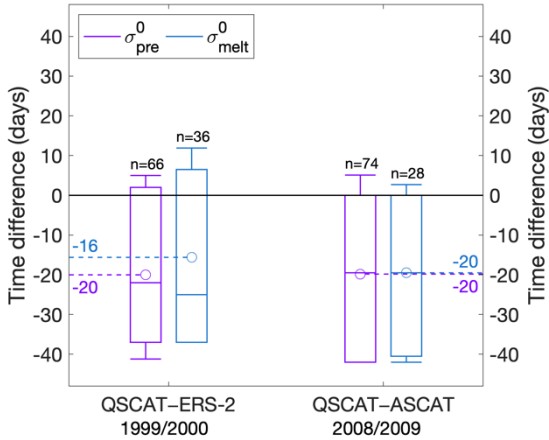

**Figure 6.** Averaged time differences between pre-melt and snowmelt onset dates retrieved from QSCAT and ERS/ASCAT at the different study locations on perennial sea ice for the overlap periods 1999/2000 and 2008/2009, respectively. Negative (positive) differences indicate an earlier (later) transition detected from Ku-band. Boxes are the first and third quartiles. Whiskers display the 20- and 80-percentiles. Circles indicate mean, lines median values.

Similar to the work of Markus et al. (2009);Mortin et al. (2014);and Stroeve et al. (2014) in the Arctic, here we used the backscatter time series of ERS-1/2, QSCAT and ASCAT to study the inter-annual variability and trends of the previously described scatterometer-derived pre-melt and snowmelt onset dates. However, as ERS/ASCAT and QSCAT use different radar bands with different penetrations depths, and have been shown to retrieve different average melt onset dates, we first quantified the differences between retrieved melt onset dates during the overlap periods of ERS-2 and QSCAT in 1999/2000 and of QSCAT and ASCAT in 2008/2009. Figure 6 shows the average time differences between Ku-band and C-band retrievals for the 12 study locations on perennial sea ice (Figure 1), for both overlap periods, respectively. On average, Ku-band QSCAT data detected pre-melt and melt onset earlier by 20 and 16 days, respectively, then the C-band ERS-2 data. Similarly, QSCAT detected earlier pre-melt and melt onset dates than ASCAT. Both, pre-melt and melt onset were observed 20 days earlier. On average, that adds up to derived pre-melt and melt onset dates from QSCAT Ku-band earlier by 20 and 18 days, respectively, than ERS and ASCAT C-band derived onset dates.

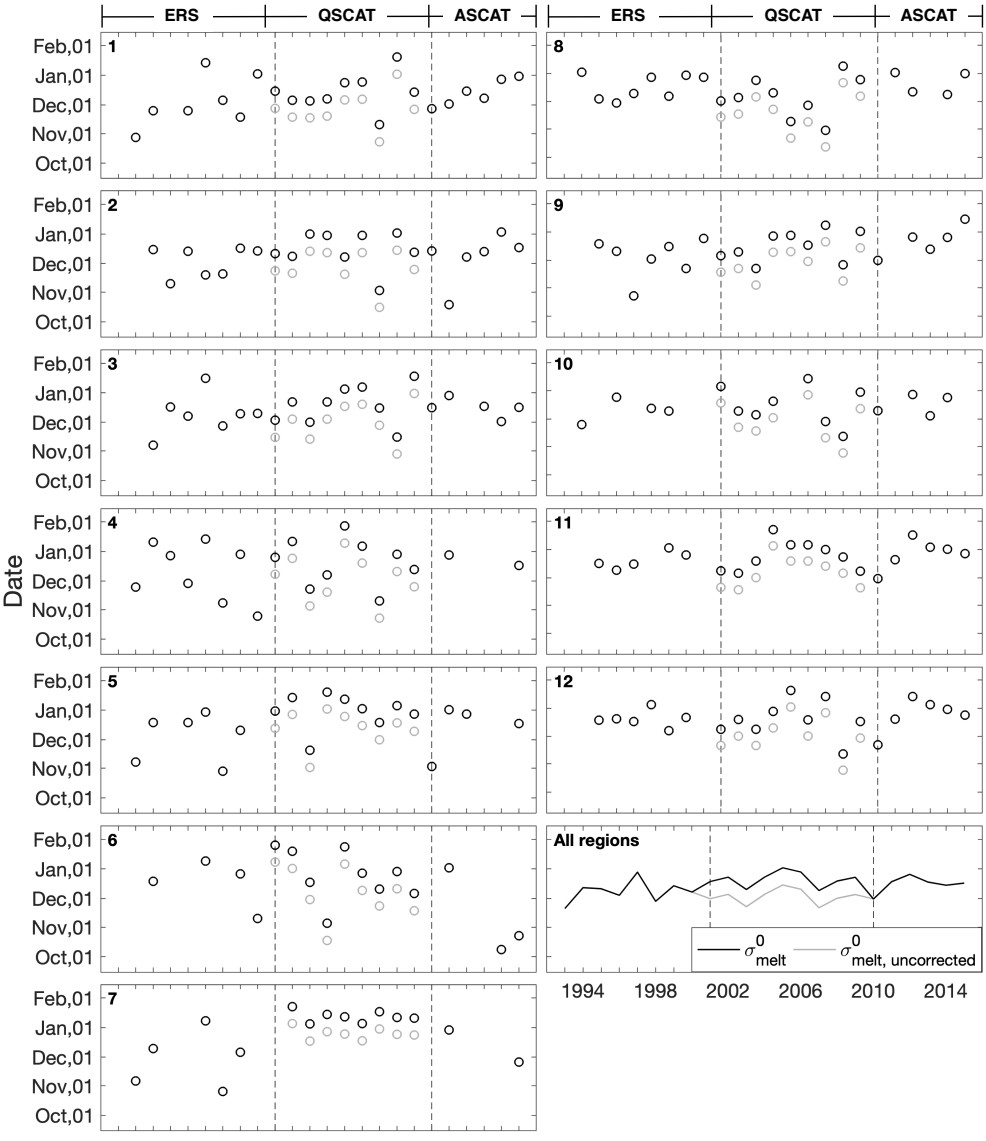

**Figure 7.** Time series of snowmelt onset dates in the perennial sea-ice zone for individual regions 1-12 (Figure 1), and all regions averaged (bottom right, upper panel). Grey circles (and grey line in bottom right) show the uncorrected QSCAT melt onset dates, while black symbols show dates corrected 11 days, the difference between Ku- and C-band melt onset dates, respectively.

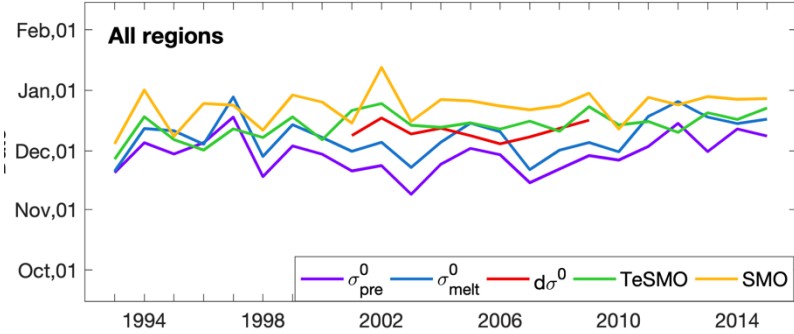

**Figure 8.** Time series of snowmelt onset dates averaged for all regions derived from backscatter time series ($\sigma^0_{melt}$, see Figure 7), diurnal backscatter variations ($d\sigma^0$), as well as the Temporary Snowmelt Onset (TeSMO) and Continuous Snowmelt Onset (SMO) derived from passive microwave observations.

Based on the results above, we corrected the QSCAT pre-melt and melt onset dates by 20 and 18 days, respectively, to derive a consistent time series of scatterometer-derived pre-melt and melt onset dates from 1992/93 to 2014/15 and to retrieve any potential trends. Figure 7 shows the resulting time series of the snowmelt onset for each study location on perennial sea ice. In addition, Figure 8 summarizes these results and also compares them with all other previously described sea ice snowmelt onset
dates from scatterometer and passive microwave observations in Antarctica. Neither single locations nor the regional averages reveal any significant temporal trend in the retrieved snowmelt onset dates. Instead, the respective snowmelt onset dates show strong interannual variations with the tendency towards later onset dates.

Figure 8 and Table 1 also show that passive microwave observed TeSMO and SMO dates occur generally later than those
observed by C-band scatterometer. More detail is provided in Figure 9 which summarizes the time differences between all observed pre-melt and melt onset dates for all locations. Results show, on average, that 68 (58) % of the pre-melt (snowmelt) onset dates retrieved from scatterometer observations occur earlier than from passive microwave observations. Particularly large negative differences are observed in the Weddell Sea, where on average about half (a quarter) of the pre-melt (snowmelt) onset differences are larger than 20 days. Overall, the mean pre-melt (snowmelt) onset difference between passive microwave
and scatterometer observations is 14 (6) days, while only 36 (40) % of the differences are smaller than 10 days, and 56 (61) % are smaller than 20 days. The smallest differences are most notably in the western Amundsen Sea (location 10), while biggest differences are detected in the southernmost Weddell Sea regions (locations 4 and 6). Again, in the Weddell Sea there is a pronounced gradient of larger differences from northwest to southeast.

Similarly, melt onset from diurnal backscatter variations $d\sigma^0$ during the QSCAT era occurred earlier than passive microwave
observed melt onset (Figure 9). However, these differences are much smaller than those from the actual time series above, with a mean (mode) or 4 (1) days.

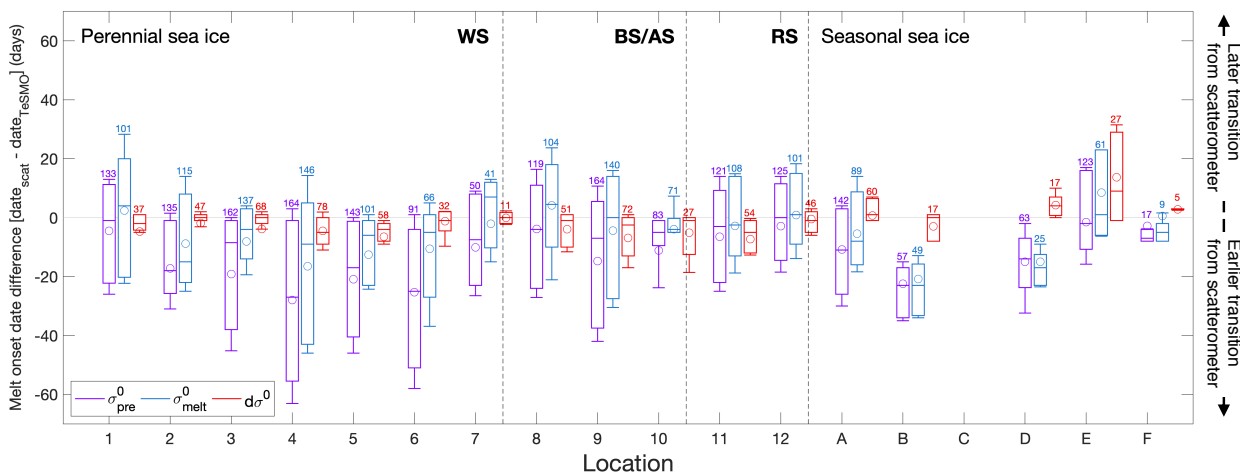

**Figure 9.** Mean time differences between retrieved snowmelt onset dates from scatterometer observations ($\sigma^0_{pre}$, $\sigma^0_{melt}$, $d\sigma^0$) and from passive microwave observations (Temporary SnowMelt Onset, according to Arndt et al. (2016)) for all study locations. Snowmelt onset dates from passive microwave observations and backscatter time series are retrieved from 1992 to 2014/15, while the retrieval of diurnal backscatter variations $d\sigma^0$ is only performed with QSCAT from 2000 to 2008. Positive (negative) differences indicate a later (earlier) transition observed by scatterometers. Boxes are the first and third quartiles. Whiskers display the 20- and 80-percentiles. Circles indicate the mean, dashes the median. Numbers above the whiskers indicate the respective sample size. The maximum sample size for derived pre-melt and melt onset dates is 207, while it is 81 only for the melt onset retrieved from diurnal variations. Abbreviations according to Figure 1: WS: Weddell Sea, BS/AS: Bellingshausen and Amundsen Seas, RS: Ross Sea.

## 4. Discussion

### 4.1 Regional differences of seasonal backscatter variations

The previously described seasonal cycle of snow processes and the related backscatter signal (Section 1) are typically found on perennial sea ice (Figures 2 and 4 a). The strong summer backscatter rise may be interrupted by few temporary backscatter drops, which can be due to local and temporary flooding or brief, strong melt events with high snow liquid water content associated with the transition towards a funicular snow regime (Figures 2 and 4 a). In the Weddell Sea section, both the magnitude of the seasonal cycle and the actual backscatter values decrease from northwest to southeast (Figure 5), with fewer melt onset dates detected in the South. This is consistent with generally colder and dryer climatic conditions further south, away from the marginal ice zone and close to the Antarctic ice sheet, where also warm air advection from the North is hampered by the Antarctic Circumpolar Trough (e.g. Simmonds and Keay, 2000;Turner et al., 2015). Consequently, both seasonal and diurnal variations in the lower atmosphere and snowpack are weak leading to less snow metamorphism and melt in the southern

part of the Antarctic sea-ice regime. The absence of distinct snowmelt processes in the southern Weddell Sea were also observed and described in previous studies on snowmelt detection from passive microwave observations (Arndt et al., 2016).

In contrast, most seasonal sea ice is not showing similarly strong seasonal backscatter cycles. Instead, the younger and thinner seasonal ice is warmer and more salty than perennial ice, and has larger brine volume at the snow/ice interface causing low backscatter through winter (Nicolaus et al., 2009;Yackel et al., 2007). Also, repeated flood-freeze cycles during winter as well as flooding events during spring and summer cause low backscatter in winter and backscatter drops during spring and summer (Drinkwater and Liu, 2000). In addition, the early break-up of seasonal sea ice associated with the formation of leads and thin ice between ice floes contributes to the declining backscatter coefficients. Moreover, due to the a comparatively large ocean heat flux of the Southern Ocean (Martinson and Iannuzzi, 1998), Antarctic sea ice might strongly melt from below and even retreat completely before the pendular-funicular transition in the snowpack have started or even dominant snow melt can develop at its surface in summer. These numerous processes coincide and compete with actual surface snowmelt processes, meltwater percolation, and superimposed ice formation, making it therefore difficult to consistently retrieve snowmelt processes and dates on seasonal sea ice by scatterometer (Section 3.2) and passive microwave algorithms (Arndt et al., 2016). We therefore consider the found snowmelt onset dates on Antarctic seasonal sea ice highly uncertain and potentially not meaningful for further analysis.

## 4.2 Inter-annual variations in the retrieved snowmelt onset dates

A main achievement of our work is the compilation of a long time series of derived snowmelt onset dates on perennial sea ice from 1992/1993 to 2014/2015, and that no significant trend in the retrieved snowmelt onset dates was found, neither from scatterometer nor from passive microwave observations. Instead we found strong inter-annual variability of all 5 snowmelt onset parameters (Table 1). However, previous studies have shown that during the same time period the duration of the summer open water season, when the ice concentration is below 15% for at least 5 days, shows significant trends in some regions. Stammerjohn et al. (2008) showed that the period of open water was longer by 85±20 days (total change from 1979 to 2004) in the Bellingshausen and Amundsen Seas (and somewhat less in the southern Weddell Sea), whereas in the perennial sea ice regime of the western Ross Sea it decreased by 60±10 days. In general, that study revealed a significant correlation between ice-season duration and sea-ice advance in most regions of the Southern Ocean, while weak correlation was found with the respective sea-ice retreat dates. The absence of similar trends for melt indicates that the timing of snowmelt and snowmelt processes are less important processes for Antarctic sea ice extent variability. Instead, previous studies have shown that seasonal and interannual ice concentration and extent variations rather depend on local oceanic conditions governing ocean heat flux and ice bottom melt, and on large-scale atmospheric circulation patterns, which affect the speed and direction of sea-ice drift and intensity of ice deformation and redistribution (Maksym et al., 2012;Stammerjohn et al., 2008;Turner et al., 2014;Turner et al., 2016).

Our melt onset retrieval algorithms utilize seasonal changes of microwave properties caused by snowpack transitions from the pendular to the funicular snow regime associated with snow metamorphism, ultimately leading to the formation of superimposed ice (Haas, 2001). Changes of these properties also affect other retrieval algorithms, of, e.g., sea-ice concentration (Willmes et al., 2014) and of ice thickness from radar altimetry (e.g. Ricker et al., 2014). For example, increasing snow metamorphism may lead to reduced radar penetration and raising elevations of radar scattering horizons, and therefore to retrievals of apparently increasing ice thicknesses. Combination of altimetric sea-ice thickness data and backscatter changes and melt processes observed by scatterometers may therefore help to improve our understanding of sea-ice surface processes and seasonal mass balance of perennial Antarctic sea ice in the future.

### 4.3 Sensitivity of different microwave wavelengths to evolving snow temperature and moisture profiles during the melt season

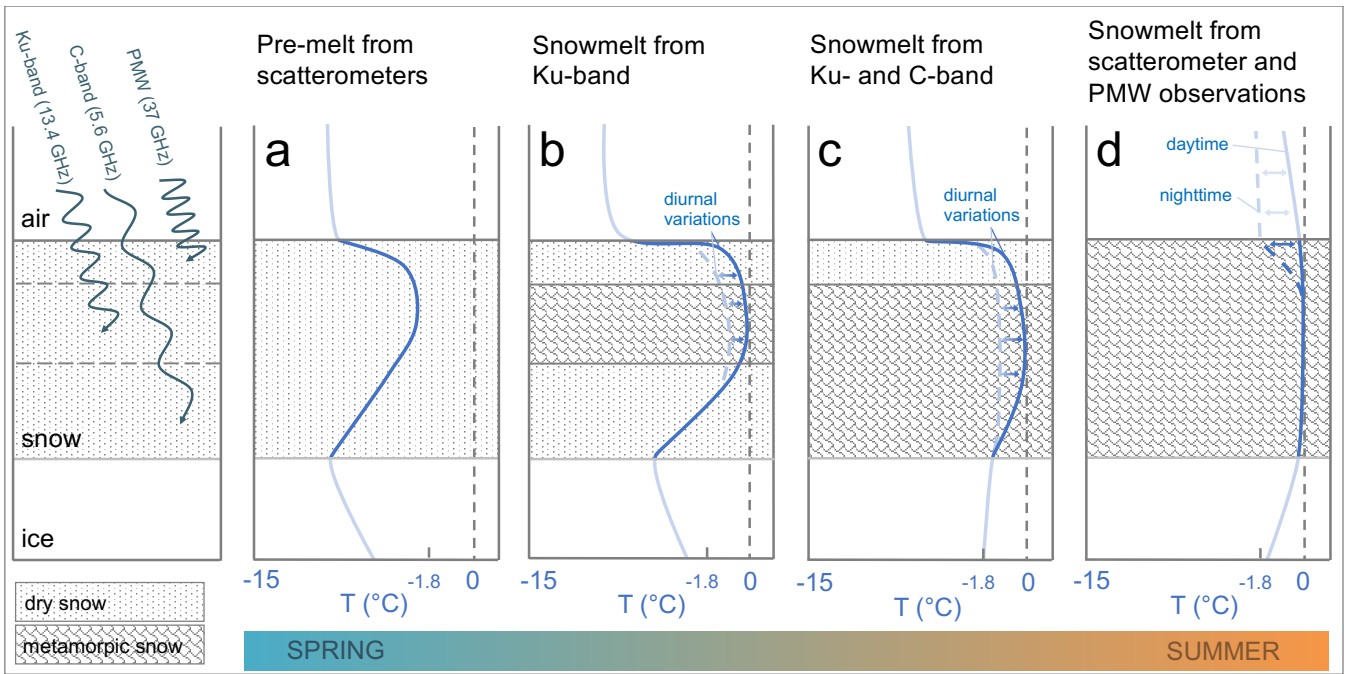

**Figure 10.** Conceptual model of the evolution of vertical air, snow and sea ice temperature profiles as well as snow metamorphism processes in the Antarctic sea ice regime during spring and summer in four characteristic stages (a-d). Note that snow temperatures close to 0°C lead to the appearance of liquid water and strong snow metamorphism. Diurnal thaw-freeze cycles during the during stage b to d cause significant differences in the snowpack properties and air temperatures between day- and nighttime. Microwaves with different wavelengths are sensitive to changing snow properties at different depths (left panel), and therefore indicate different melt-onset dates in the same region.

An important result of this study is that the retrieved perennial ice snowmelt onset dates from Ku-band (QSCAT), C-band (ERS and ASCAT), and 37 GHz passive microwaves (PMW) were different. On average, melt was detected first with Ku-band, followed by C-band (20 and 18 days later on average), and then PMW (14 and 6 days later on average), see Figures 6, 8, 9, and Table 1). This behavior could be related to the different wavelengths of those three observations with their different penetrations depths, and different sensitivity to increasing Rayleigh scattering due to metamorphic processes with cycles of variable liquid water content, growing snow grains and ice layer formation (e.g. Onstott, 1992;Ulaby et al., 1986). However, the observed behavior indicates that medium wavelength Ku-band signals with their specific wavelength sense these processes first, followed by longer wavelength C-band signals with their deeper penetration, and that short-wavelength PMW signals with the least penetration are affected last.

We therefore propose a conceptional model for the temporal evolution of snow column temperature, moisture, and metamorphism profiles which can qualitatively explain the observed microwave behavior and which suggests that depth-dependent changes of snow properties can be retrieved by multi-frequency satellite microwave observations. The model is illustrated in Figure 10.

During the spring and pre-melt phase, subject to strong diurnal variations, the snow and ice warm from above due to atmospheric turbulent and shortwave radiative heating, with the highest temperatures in the upper, interior snow layers while lower layers remain cooler due to their vicinity to the cold, underlying sea ice with its large heat capacity (Cheng et al., 2003;Haas et al., 2001). During this time, brine which may potentially reside within the snow pack due to brine wicking or sea spray deposition during winter is removed from the snow pack by downward percolation, leading to the fact that snow salinity on sea ice is generally found to low in summer (Massom et al., 2001; Haas et al., 2001; Nicolaus et al., 2009). Note that the presence of brine lowers the melting temperature at snow grain boundaries, and therefore brine remains liquid at temperatures well below 0°C and can percolate downwards before the time of actual melt onset.

As the snow warms and reaches near melting temperatures internally, at the very snow surface temperatures can remain lower than in the interior due to longwave radiative cooling of the surface. This so-called solid-state greenhouse effect can lead to a sub-surface temperature maximum (Figure 10 a) that can eventually lead to subsurface melt. Such internal melting is frequently observed in snow and blue ice near the coast of Antarctica, and the effect has been calculated by, e.g., Cheng et al. (2003);Liston and Winther (2005);and Liston et al. (1999). The magnitude of the difference between the snow surface temperature and the sub-surface temperature is debated, and depends on snow density, grain size, and ambient atmospheric conditions (Brandt and Warren, 1993). However, the dense snow, large grains with low specific surface area (SSA), and the presence of ice layers all typical for perennial Antarctic sea ice (Nicolaus et al., 2009) support the occurrence of sub-surface melt, as does the snow and ice's presence at sea level and at more northern latitudes than most of the coast of Antarctica (Liston and Winther, 2005).

As the snow warming continues and successively affects lower and lower snow layers, first the interior and then lower layers reach near-melting or sporadic melting temperatures (Figure 10 b and c) with increases of liquid water content and diurnal or multi-day transitions into the funicular snow regime, when they are subject to strong, irreversible snow metamorphism and ice lenses or superimposed ice formation (Cheng et al., 2003;Haas et al., 2001). Only at a later stage, during instances of large

atmospheric heat fluxes, the very snow surface becomes warm enough to be subjected to strong snow metamorphism as well (Figure 10 d).

The different stages in Figure 10 a-d are frequently interrupted or ending by colder periods which cause the prominent thaw/refreeze cycles. While the appearance of liquid water will generally reduce radar backscatter and increase microwave emissivity, it is the refreezing of the respective snow layers and irreversible snow metamorphism that causes the backscatter increases utilized in this paper.

With their intermediate penetration, it is therefore plausible that Ku-band signals sense initial snow property changes in the interior snow column first (Figure 10 b), while C-band and PMW signals receive their strongest contributions from the unchanged lower and topmost layers and therefore show no response at that time. C-band signals respond next, when the warming and sporadic wetting has reached the lower layers near the snow/ice interface (Figure 10 c). Finally, PMW signals, receiving their main power from the topmost snow layer, are only affected once the very surface experiences thaw-refreeze cycles as well.

It should be noted that the penetration depth of both Ku- and C band into dry snow is at least more than a meter (Ulaby et al., 1986). However, increased backscatter along the propagation path through the snow at any depth will result in the observed overall backscatter increases. The longer wavelengths of C-band signals may not be sensitive to the initial metamorphism in the interior snow layers which is already sensed by the shorter Ku-band signals (Figure 10 b). Overall, at the 12 study locations on perennial ice the mean amplitude between the seasonal minimum and maximum is 13.35dB for the QSCAT time period, but only 7.66 and 9.35dB for ERS and ASCAT, respectively (Figure 2). This also indicates that Ku-band backscatter could be more strongly affected by a certain degree of snow metamorphism than C-band backscatter.

The mean time differences of 18 days between melt onset detected by Ku-band and C-band, and 6 days between C-band and PMW provides a reference for the time scales of initial warming and snow metamorphism in the Antarctic. Interestingly, in the Arctic Mortin et al. (2014) found hardly any temporal difference between melt onset dates observed by the same sensors. In agreement with our conceptual model above we interpret that behavior by the fact that snow on Arctic sea ice usually warms very rapidly throughout the entire snow column during melt-onset when it is already close to the melting temperature, and therefore all wavelengths respond at approximately the same time. This is consistent with the generally very rapid melt and disappearance of snow on sea ice in the Arctic discussed in the introduction.

## 5. Summary and conclusions

In this study, we compiled a time series of snowmelt onset dates on Antarctic sea ice from 1992 onwards using different radar scatterometer observations (ERS-1/2, ASCAT: 5.3 GHz; QSCAT: 13.4 GHz), in extension of previous work by Haas (2001). Doing so, we defined two major snowmelt stages: Pre-melt onset associated with the initial warming and increasing appearance of liquid water in the snowpack, followed by snowmelt onset related to diurnal thawing and refreezing of the snowpack. Results show that the magnitude of seasonal and diurnal backscatter variations is highly dependent on latitude, related to earlier and more frequent snowmelt in the north (mid of November in the northern Weddell Sea). In contrast, regions farther south

and closer to the continent reveal weaker seasonal and diurnal backscatter variations, and therefore less snowmelt attributed to the prevalent atmospheric conditions of cold and dry air and the absence of warm air advection events from the North.

For practical reasons we were unable to update this time series to include the 2015/16 to 2018/19 summer seasons, when Antarctic sea ice extent in September had reached record minima (e.g. Schlosser et al., 2018). It would be interesting to see in

a follow up study if significantly earlier melt onset dates would be observed during those recent years. However, we believe that they will not, based on the facts that previous changes of ice extent did not strongly affect melt onset dates as mentioned above, that those recent changes may have a strong contribution from oceanic processes (Gordon et al., 2007;Turner et al., 2015), and that most potential surface changes may occur in the seasonal ice zone or closer to the marginal ice zone where our algorithm is not applicable.

Based on the observed successive timing of melt events retrieved from different sensors, i.e. C- and Ku-band radars and 37 GHz passive microwave radiometers, we developed a conceptual model of the temporal evolution of temperature and metamorphism of thick snow on perennial Antarctic sea ice and their effect on, and detectability by, different microwave sensors during the spring/summer transition: The model explains qualitatively how snow metamorphism occurs first in interior snow layers and mostly affects Ku-band signals. Once warming has reached the lower snow pack the resulting metamorphism

there can be detected by C-band signals. The topmost snow layer remains coldest the longest by radiative cooling. Only when it warms and is affected by thaw/refreeze cycles at last do short wavelength passive microwave signals respond as well.

Based on the results obtained here, we suggest that there is a potential to observe snow processes at different depths from space, opening new avenues for studies of energy and mass budgets of snow on sea ice in the Southern Ocean. In addition, improved observations of snow metamorphism will contribute to better understanding and correction of uncertainties and

spatial variability of space-borne retrievals of sea-ice concentration, snow depth and sea-ice thickness. However, improved microwave modeling informed by new in-situ observations of microwave properties and snow melt processes on Antarctic sea ice are required first to inform such activities.

**Acknowledgement**

Satellite data of radar backscatter were kindly provided by the Scatterometer Climate Record Pathfinder (SCP) project,

sponsored by NASA (http://www.scp.byu.edu/), and sea-ice concentration data from the NASA National Snow and Ice Data Center Distributed Active Archive Center, Boulder, Colorado, USA. We acknowledge discussions about microwave emission and scattering behavior with Wolfgang Dierking, Sascha Willmes, Marcus Huntemann, and during meetings within the ISSI project "Satellite-derived estimates of Antarctic snow- and ice-thickness". The work was funded by the Helmholtz Alliance "Remote Sensing and Earth System Dynamics" (HA-310) and the Alfred-Wegener-Institut Helmhotz-Zentrum für Polar- und

Meeresforschung, Bremerhaven, Germany. The retrieved snowmelt onset dates from scatterometer data observations are available at PANGEA.

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

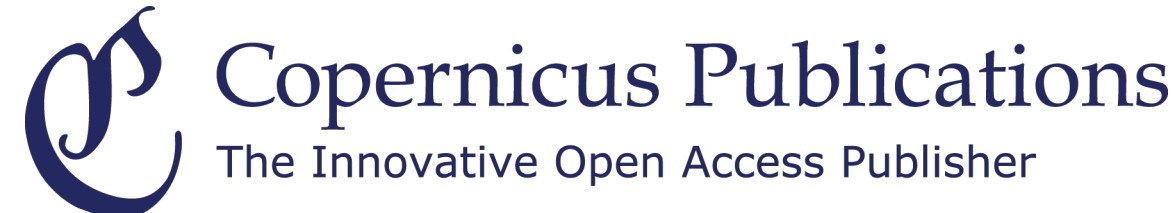

**Figure 1: The logo of Copernicus Publications.**