# Peer review of "Spatio-temporal variability and decadal trends of snowmelt processes on Antarctic sea ice observed by satellite scatterometers"

_The Cryosphere, 2019_

## Referee Comment (RC1) · Anonymous Referee #1 · 10 Mar 2019

The manuscript builds on the study of Haas et al. (2001) on the analysis of spaceborne scatterometer data for investigating snow melt. The work compiles a time-series of melt onset information from ERS-1/2, QuikSCAT, and ASCAT for the 1992 to 2015 period. The retrieved dates are then compared to those derived from passive microwave data. The analysis presents new and relevant information about snow on Antarctic sea ice and the capability to remotely sense snow conditions on Antarctic sea ice. There are several aspects of the analysis that require clarification, more details and revised inter-pretation based on previous studies. Please find detailed comments below.

[Figure]

Page 1, line 6. From a broader view, the presence of snow would be the key driver, wouldn't it?

Page 1, line 13. Metamorphism is an umbrella term for several types of metamorphic processes of snow. What exactly do the authors mean by metamorphism? The use of it in the manuscript is ambiguous at times. I recommend adding "melt" or "freeze-melt" before "metamorphism" throughout the manuscript to differentiate it from other metamorphic processes.

Page 1, lines 23-26. I suggest rewriting this section since it's an over-extension of the results. The conceptual model hypothesizes that the evolution of seasonal snow temperature profiles could affect different microwave bands. It's also important to define the environmental conditions that the conceptual model is limited to. From what's been described, it seems like the conceptual model would be applicable to a snowpack with low density, no damp/saline basal layer, no internal icy layers or lenses, somewhat uniform vertical grain structure, and over perennial ice where high-frequency diurnal temperature fluctuations occur simultaneously with a slow, steady increase in mean temperature.

Page 2, line 10 and throughout the manuscript. Please specify which Haas et al. 2001 paper that you refer to.

Page 2, line 10. How do variations in snow properties affect the mass budget of the ocean?

Page 2+ Snowmelt onset in the Antarctic is described as more subtle than the Arctic. What is meant by subtle? Previous studies describe warm, marine cyclones that bring dramatic temperature swings and/or rainfall throughout the year, and with increasing frequency going into summer. The induced melt from these events is not subtle and typically results in a more structurally-complex snowpack. Similarly, the manuscript overly generalizes Arctic snowmelt, as on page 21 line 9-10, "...warms very rapidly throughout the entire snow column during melt onset..." What observations support

these statements? Snowmelt is often not as rapid and continuous in the Arctic as described in the manuscript. Peng et al. 2018 is an insightful example with their definitions of melt onset periods. Studies have shown numerous snowfall and freeze events during spring and summer, which highlights the discontinuous nature of melt (and freeze) that seems to occur in all snow environments.

Page 2, line 31. In contrast to what?

Page 2+. Salinity affects radar backscatter. Previous observations not only show brine wicking up to 15-20 cm into the Antarctic snowpack from its base, but that as a whole the Antarctic snow cover is saline. I encourage the authors to consider the effects of salinity on the retrieved dates and adding in a discussion on this topic in the manuscript.

Page 3, line 2. Arndt et al. 2016 seems like the wrong reference here.

Page 3, lines 17-18. If flooding is indeed an important mechanism for Antarctic snow and sea ice as described in Massom et al. (2001), I recommend adding more discussion on what the potential effects of flooding are on the results. Here and elsewhere in the manuscript, flooding is swept "under the rug" so to speak by suggesting that it only occurs right before the sea ice cover disintegrates or is limited to the edge of the sea ice pack.

Page 3, line 23. Superimposed ice. Do we know that this it is a wide-spread phenomenon during snow melt onset? My understanding is that a substantial amount of snowmelt is required before the meltwater can fully percolate down to the ice surface. I suspect superimposed ice would occur after snow melt onset for this reason. The observations in Haas et al. 2001 were ~2 months after the snow melt onset dates shown here, so it's not clear if the presence of superimposed ice can be used to interpret the backscatter for identifying melt onset. There may be comparable situations in the Antarctic where superimposed ice does not form at all, see Polashenski et al. 2017 for an Arctic example. Based on the literature, rainfall may also be important to consider in Antarctic snow.

Page 4, line 3. "Adjusted" would be a more appropriate word that "corrected" here and elsewhere in the manuscript.

Page 7. The sample size is limited to a pixel for each location to reduce the variability associated with different ice conditions. How sensitive are the results to one pixel vs. a multi-pixel average? I would suspect that variability is larger for a single pixel due to the advection of ice with differing properties. An eight-neighbor mean may be more stable.

Page 7, line 10. It would be helpful to clarify that the sea ice concentrations are from the Bootstrap algorithm.

Page 5, lines 15-16. What information was used to determine which areas were predominantly seasonal and perennial ice? Is there the possibility that some years had a mixture of ice types at the designated sites?

Page 5, lines 18. Anderson, Bliss, Peng appear to be under-referenced with regard to melt onset detection from passive microwave data.

Page 7, lines 26-29. How would a 70% sea ice concentration threshold remove flooded ice from your sampling? How are ice concentration and flooded sea ice related?

Page 8, figure 4. It would be helpful to show the sea ice concentration here and either as additional figures or in supplementary information for figures 2 and 3 given its influence on backscatter. How were the start and end points of the bolded solid lines determined?

Page 9, lines 1-11. How much of this is speculation? Were coincident in situ observations linked with observed changes in backscatter? Please clarify in the manuscript.

Page 9, line 10-11. Please specify that you mean a positive albedo feedback. The manuscript neglects here and elsewhere the possibility of stopping surface melt due to fresh snowfall.

Page 9, line 20-22. How was October 1st determined? How were the 2 dB and 3 dB thresholds determined? Are the results sensitive to these choices?

Page 9, lines 26-30. What fraction of the time-series had indeterminable melt dates for perennial and seasonal ice? It would be helpful to put those numbers in the results section.

Page 10, line 5. It would be helpful to give more detail on what the "regionally adaptive" approach does.

Page 10, lines 18-20. It would be helpful to give more detail here. What is the iterative algorithm converging on exactly? Is a priori information on thresholds needed?

Page 12, line 14. Is 7.66 dB different from the value found in Haas et al. 2001? If so, why?

Page 13, Section 3.2. Similar to an earlier comment, approximately what fraction of the time-series had detectable melt onset? This can help provide the reader with context on the limitations (and possibilities) of this approach over seasonal ice.

Page 17, figure 9. It would be helpful to give the sample size of each mean difference, either in the figure or in a table, so that readers can appropriately interpret the spread.

Page 18, lines 1-3. I suggest rewording this sentence. As it's stated, it sounds like perennial ice has larger brine volume at the surface, which is probably not what you mean.

Page 18, lines 7-9. Do you have a reference for this statement?

Page 18, line 25. "Instead, we suggest…" I recommend changing this to: "Instead, other studies have shown…" since this analysis does not show results on these topics.

Page 19, line 3. "…seasonal mass balance of Antarctic sea ice in the future." Based on the results, isn't this approach only appropriate for perennial sea ice? If so, it would be good to make that clarification in this ending paragraph.

Page 19, Section 4.3. This is an interesting idea, but it misses some fundamental characteristics of snow, the most significant being light penetration in snow vs. blue ice, the existence of a saline, damp layer at the base of the snowpack and icy layers and lenses within the snowpack. The Brandt and Warren (1993) study shows that visible wavelengths are not absorbed at depth in a snowpack, but are scattered back to the surface. Near-infrared wavelengths only get absorbed in the top few millimeters of the snowpack. The study then describes optimal conditions where sub-surface melt could be important. These are low-albedo ice like blue ice and low-density snow, like depth hoar. Both conditions are not typical of snow on Antarctic sea ice. The description on page 20, lines 11-16 must be a misinterpretation of the Brant and Warren analysis and needs revising. Related, several studies have since shown technical issues with radiative heating of sensors, such as in Cheng et al. 2003, making observed sub-surface temperature increases somewhat dubious. Secondly, there is a wealth of papers that show the widespread occurrence of icy layers and a damp, saline basal layer in snow on Antarctic sea ice, in contrast to an assumption of dry snow as on page 20, line 33. This damp layer, as well as internal icy layers within the snowpack, greatly modify the electromagnetic signature of snow, its temperature gradient and snow metamorphism. I encourage the authors to give these aspects consideration and incorporate them into the proposed hypothesis. If the hypothesis in Section 4.3 conflicts with typical characteristics of Antarctic snow, then explicitly state that it does and describe specifically which environmental conditions the hypothesis is limited to. Although simple, the schematic in Garrity (1992) is informative and may help with this. For figure 10, it would be helpful to overlay snow grain symbols so that readers can have a better idea of which melt-induced characteristics you're referring to in the snowpack for each stage. The WMO and Colbeck (1991) would be useful references for this.

Page 21, lines 24-27. Could you provide some references to support these statements? Also, what is meant by "the most potential surface changes?"

Page 21-22, lines 30-32/lines 1-2. How do we know this is true? Are there references

to support these statements?

Page 22, line 3. It's stated that the results obtained in this study demonstrate the potential to observe snow processes at different depths from space. This is not true. However, the study does hypothesize that this could be possible, which is different from a demonstration. Please correct this for clarity.

————————————————

---

## Referee Comment (RC2) · Anonymous Referee #2 · 11 Mar 2019

General Comments:

The authors examined the onset of snow melt over Antarctic sea ice using data sets from scatterometers (ERS-1/2, QSCAT and ASCAT) and passive microwave radiometers. Between 1992 and 2015, they found insignificant changes in onset dates which they claim be consistent with the small trends in Antarctic sea ice extent. Also, they used the differential lag in onset timing between the observing instrument to develop a conceptual model for inferring the evolution of the depth/temperature-dependent snow processes during the onset period, and conclude that multi-wavelength instruments

may be able to provide information on the behavior of the snow column on Antarctic sea ice during early melt.

Throughout the paper, the authors contrasted the behavior of the Arctic and Antarctic – I find those discussions to be interesting and useful.

The only comment is that the authors inferred from only a few samples (12) the general behavior of the onset-dates of circumpolar ice cover and their relationship to the observed trend in ice extent. This is less credible without more justification as to why a non-uniform sampling of the ice cover (Fig. 1) is sufficient for this analysis.

I would like the comments below addressed prior to publication.

Detailed Comments:

Page:Line number

1:28 While it is true that the circumpolar ice extent has changed insignificantly over the period of study, the trends are significant in the different sectors (e.g., Ross Sea sector). I think that fact should be noted and there are implications as far as the discussions in the remainder of the text regarding expected trends in the onset dates and ice extent.

3:20 This is in contrast to what is expected in the Arctic, where backscatter from perennial ice is expected to decrease during the summer. Perhaps another point to note.

6:19 Need clarification: Is it the daily product that was used or the twice daily product? On p.6, the text indicated only the daily product is used.

7:10 Please specify which ice concentration product is used here.

7:15 Perhaps it's good to point out how the samples were 'carefully' chosen to reflect/represent the large-scale behavior/trends of the Southern Ocean ice cover.

9:1 I don't think 'extensive' is appropriate here – let the reader decide.

Figures 3&4. Shouldn't the seasonal ice in sample D disappear during the summer?

Figure 5. Are the backscatter for both C-band and Ku-band merged here? If so, please indicate so because one would expect differences between the two wavelengths.

18:8 Large oceanic heat flux (I imagine relative to the Arctic) – a reference is appropriate here.

18:13 meaningful for? For indicating the ice extent?

18:24 I think this is what you are referring to above, i.e., 18:13.

Section 4.2 the suggestion here that snow processes may be secondary in explaining the ice extent is important – perhaps worth noting in the abstract. Your work points to that and it is geophysically important, but saying that it is 'NOT' important may be a bit too strong without more discussion and supporting evidence.

20:14 This conceptual model depends on the initial changes to occur in the subsurface prior to that on the surface such that permittivity changes in the interior, while in the pendular regime, leads the change at the surface. I think this is perhaps too dependent on the temperature argument. Is the temperature profile entirely necessary?

---

## Author Comment (AC1) · 2 Apr 2019

The manuscript builds on the study of Haas et al. (2001) on the analysis of spaceborne scatterometer data for investigating snow melt. The work compiles a time-series of melt onset information from ERS-1/2, QuikSCAT, and ASCAT for the 1992 to 2015 period. The retrieved dates are then compared to those derived from passive microwave data. The analysis presents new and relevant information about snow on Antarctic sea ice and the capability to remotely sense snow conditions on Antarctic sea ice. There are several aspects of the analysis that require clarification, more details and revised interpretation based on previous studies. Please find detailed comments below.

We highly appreciate the great work that the reviewer put into revising our manuscript. This work is really excellent and we realize that he / she is really familiar with this field of research and has a great expertise to make most useful comments and suggestions. We thank the reviewer for the very critical questioning regarding the described relevant and dominant processes causing seasonal changes in the properties of the Antarctic snowpack. To overcome these missunderstandings, we added more explanations in the respective sections as pointed out in the following explicite responses.

Page 1, line 13. Metamorphism is an umbrella term for several types of metamorphic processes of snow. What exactly do the authors mean by metamorphism? The use of it in the manuscript is ambiguous at times. I recommend adding "melt" or "freeze- melt" before "metamorphism" throughout the manuscript to differentiate it from other metamorphic processes.
We have added "destructive" to describe what kind of metamorphism we refer to. Other than that we believe that it is clear from the context that this is what is meant, and don't feel that it needs to be repeated frequently.

Page 1, lines 23-26. I suggest rewriting this section since it's an over-extension of the results. The conceptual model hypothesizes that the evolution of seasonal snow temperature profiles could affect different microwave bands. It's also important to define the environmental conditions that the conceptual model is limited to. From what's been described, it seems like the conceptual model would be applicable to a snowpack with low density, no damp/saline basal layer, no internal icy layers or lenses, somewhat uniform vertical grain structure, and over perennial ice where high-frequency diurnal temperature fluctuations occur simultaneously with a slow, steady increase in mean temperature.
We have added "on thick Antarctic snow" to the abstract to indicate that our model may not be applicable to all sea ice regions worldwide. Other than that we don't feel that much more information should be given in the abstract., although we have added more specific information later in the text.

Page 2, line 10 and throughout the manuscript. Please specify which Haas et al. 2001 paper that you refer to.
Throughout the manuscript as well as in the reference list we carefully distinguish between Haas et al, 2001 and Haas, 2001.

Page 2, line 10. How do variations in snow properties affect the mass budget of the ocean?
We agree that the given sentence is misleading and therefore adapted it towards the "mass budget of sea ice".

Page 2+ Snowmelt onset in the Antarctic is described as more subtle than the Arctic. What is meant by subtle? Previous studies describe warm, marine cyclones that bring dramatic temperature swings and/or rainfall throughout the year, and with increasing frequency going into summer. The induced melt from these events is not subtle and typically results in a more structurally-complex snowpack. Similarly, the manuscript overly generalizes Arctic snowmelt, as on page 21 line 9-10, "...warms very rapidly throughout the entire snow column during melt onset..." What observations support these statements? Snowmelt is often not as rapid and continuous in the Arctic as de- scribed in the manuscript. Peng et al. 2018 is an insightful example with their definitions of melt onset periods. Studies have shown numerous snowfall and freeze events during spring and summer, which highlights the discontinuous nature of melt (and freeze) that seems to occur in all snow environments.
Which studies does the reviewer refer to? Our statements are meant in general and address the fact that no melt ponds are observed in the Antarctic, in contrast to the Arctic. It is clear that there are instances when melt in the Arctic is retarded by new snow events or cold spells. However, this does not change the general recognition that snow melt in the Antarctic is much more subtle/slow/sporadic than in the Arctic. A more detailed deicussion of differences between Arctic and Antarctic snow processes is beyond the scope of that paper. We have clarified the text:

However, on Antarctic sea ice, the retrieval of snowmelt onset is more challenging because thawing and melting are weaker and more sporadic than in the Arctic. There is widespread occurrence of diurnal thaw-freeze cycles (Haas et al., 2001;Nicolaus et al., 2006;Nicolaus et al., 2009), or the snow may only thaw during the passage of warm marine cyclones, with the snow refreezing shortly after (Willmes et al.,xxx). These thaw-refreeze events cause strong,

destructive snow metamorphism. Under more intensive melting conditions, snow changes from the pendular to the funicular regime (e.g. Denoth, 1980) where the liquid snow melt water percolates through the snowpack to lower, colder layers or to the ice surface where it refreezes to form superimposed ice (Tison et al., 2008;Haas et al., 2008;Haas et al., 2001;Nicolaus et al., 2009;Willmes et al., 2009).

Page 2, line 31. In contrast to what?
'In contrast" was meant to distinguish between dominant processes in winter (snow-ice formation) and spring/summer (superimposed ice). However, to reduce the confusion, we removed the discussion of flooding and snow ice from this section and have included it in our later discussion of winter processes preceding the transformation of first-year ice to perennial ice during melt onset.

Page 2+. Salinity affects radar backscatter. Previous observations not only show brine wicking up to 15-20 cm into the Antarctic snowpack from its base, but that as a whole the Antarctic snow cover is saline. I encourage the authors to consider the effects of salinity on the retrieved dates and adding in a discussion on this topic in the manuscript.
We are well aware of the saline nature of some snow on first year ice, and that there can be widespread flooding. These properties are also responsible for the ice's low backscatter in winter. However, once the snow warms the brine typically drains and snow salinity measurements in summer show negligible salinity throughout. The text has been rewritten significantly to clarify these points.

Page 3, line 2. Arndt et al. 2016 seems like the wrong reference here.
We agree and deleted it.

Page 3, lines 17-18. If flooding is indeed an important mechanism for Antarctic snow and sea ice as described in Massom et al. (2001), I recommend adding more discussion on what the potential effects of flooding are on the results. Here and elsewhere in the manuscript, flooding is swept "under the rug" so to speak by suggesting that it only occurs right before the sea ice cover disintegrates or is limited to the edge of the sea ice pack.
See above, we have consolidated the text to better separate between winter and summer properties and processes, and have described that salty snow and flooding are responsible for low backscatter during winter. We have then clarified that the snow desalinates during spring and that negative freeboard is uncommon on perennial ice.

Page 3, line 23. Superimposed ice. Do we know that this it is a wide-spread phenomenon during snow melt onset? My understanding is that a substantial amount of snowmelt is required before the meltwater can fully percolate down to the ice surface. I suspect superimposed ice would occur after snow melt onset for this reason. The observations in Haas et al. 2001 were ~2 months after the snow melt onset dates shown here, so it's not clear if the presence of superimposed ice can be used to interpret the backscatter for identifying melt onset. There may be comparable situations in the Antarctic where superimposed ice does not form at all, see Polashenski et al. 2017 for an Arctic example. Based on the literature, rainfall may also be important to consider in Antarctic snow.
We agree with the reviewer that superimposed ice is not responsible for the initial backscatter rises. However, together with icy snow and ice layers it contributes to maintaining high radar backscatter by the end of the summer. We have clarified this in the text by restructuring and adding a few words. We added a few words about passage of warm cyclones to include cases of rain fall. Again, these events are mostly sporadic and typically lead not to strong, accelerated melt nor the formation of melt ponds.

Page 4, line 3. "Adjusted" would be a more appropriate word that "corrected" here and elsewhere in the manuscript.
We agree and adapted it.

Page 7. The sample size is limited to a pixel for each location to reduce the variability associated with different ice conditions. How sensitive are the results to one pixel vs. a multi-pixel average? I would suspect that variability is larger for a single pixel due to the advection of ice with differing properties. An eight-neighbor mean may be more stable.
We had actually conducted our analysis for individual pixels and groups of 3x3 pixels. We agree that a larger region of 3x3 pixels will provide more representative results for the respective areas. Therefore, we are now reporting the results of the analysis of the 3x3 pixel regions. However, this has no effect on the presented results in the manuscript but the given dates shifted slightly by 1-2 days back or forward.

Page 7, line 10. It would be helpful to clarify that the sea ice concentrations are from the Bootstrap algorithm.
We agree and added that information.

Page 5, lines 15-16. What information was used to determine which areas were pre- dominantly seasonal and perennial ice? Is there the possibility that some years had a mixture of ice types at the designated sites?
The given locations were chosen to agree with those of Haas (2001) in order to ensure both a proper comparison between both studies and a reasonable continuation of the given time series. However, it can not be ruled out that single

points are not in all given years covered by a MYI regime. In such years there may not be any results because our algorithm does not work for thin, deteriorating FYI. We have stated the percentage of successful retrievals on page xxx: Overall, pre-melt and melt onset dates were retrievable for 79% and 64% of all analyzed pixels. For the seasonal sea-ice regime, pre-melt and melt onsets were obtained for 46% and 26% of the analyzed pixels.

Page 5, lines 18. Anderson, Bliss, Peng appear to be under-referenced with regard to melt onset detection from passive microwave data.
Thank you for these suggestions but we prefer to limit our citations to a few, most relevant and recent publications.

Page 7, lines 26-29. How would a 70% sea ice concentration threshold remove flooded ice from your sampling? How are ice concentration and flooded sea ice related?
Clarified: This avoided contamination of results by wind-roughened water (Drinkwater and Liu, 2000), and effectively eliminated regions of deteriorating, thin ice where surface flooding and break up into small floes and brash ice may occur, e.g. in the marginal ice zone, with competing effects on backscatter evolution.

Page 8, figure 4. It would be helpful to show the sea ice concentration here and either as additional figures or in supplementary information for figures 2 and 3 given its influence on backscatter. How were the start and end points of the bolded solid lines determined?
For Figure 4 the grey shaded area indicates the part of the time series with sea-ice concentration less than 70%. As given in the figure caption, the bold lines indicate the time period included in the transition retrieval (01 Oct to 31 January). However, we added these dates in order to make this clearer.

Page 9, lines 1-11. How much of this is speculation? Were coincident in situ observations linked with observed changes in backscatter? Please clarify in the manuscript.
Most of our observations were carried out during the ISPOL drift station in 2004/05. We added more references to that part.

Page 9, line 10-11. Please specify that you mean a positive albedo feedback. The manuscript neglects here and elsewhere the possibility of stopping surface melt due to fresh snowfall.
We agree and clarified that.

Page 9, line 20-22. How was October 1st determined? How were the 2 dB and 3 dB thresholds determined? Are the results sensitive to these choices?
The thresholds are based on the average rise of the backscatter values during the spring/summer transition. This is stated in the last sentence of the paragraph. It is unlikely that melt onset would occur before.

Page 9, lines 26-30. What fraction of the time-series had indeterminable melt dates for perennial and seasonal ice? It would be helpful to put those numbers in the results section.
We agree and added the percentage of the respective successful retrievals: Overall, pre-melt and melt onset dates were retrievable for 79% and 64% of all analyzed pixels. For the seasonal sea-ice regime, pre-melt and melt onsets were obtained for 46% and 26% of the analyzed pixels.

Page 10, line 5. It would be helpful to give more detail on what the "regionally adaptive" approach does.
The working principle of the regionally adoptive approach is described right after the approach is mentioned.

Page 10, lines 18-20. It would be helpful to give more detail here. What is the iterative algorithm converging on exactly? Is a priori information on thresholds needed?
The iterative threshold selection algorithm derives from the given $d\sigma^0$ values per pixel the optimized threshold in order to distinguish between summer and winter conditions. We slightly adjusted that sentence in order to make this clearer.

Page 12, line 14. Is 7.66 dB different from the value found in Haas et al. 2001? If so, why?
Numbers are slightly different from the ones given in Haas (2001) as the amount of analyzed pixels slightly differ between both studies.

Page 13, Section 3.2. Similar to an earlier comment, approximately what fraction of the time-series had detectable melt onset? This can help provide the reader with context on the limitations (and possibilities) of this approach over seasonal ice.
We agree and provided the respective fraction of detectable dates.

Page 17, figure 9. It would be helpful to give the sample size of each mean difference, either in the figure or in a table, so that readers can appropriately interpret the spread.

We agree and added these numbers to the figure.

Page 18, lines 1-3. I suggest rewording this sentence. As it's stated, it sounds like perennial ice has larger brine volume at the surface, which is probably not what you mean.

Reworded: Instead, the younger and thinner seasonal ice is warmer and more salty than perennial ice, and has larger brine volume at the snow/ice interface causing low backscatter through winter

Page 18, lines 7-9. Do you have a reference for this statement?

Yes, we added a respective reference: (Martinson and Iannuzzi, 1998).

Page 18, line 25. "Instead, we suggest. . ." I recommend changing this to: "Instead, other studies have shown. . ." since this analysis does not show results on these topics.

We agree and added that.

Page 19, line 3. ". . .seasonal mass balance of Antarctic sea ice in the future." Based on the results, isn't this approach only appropriate for perennial sea ice? If so, it would be good to make that clarification in this ending paragraph.

We agree and specified that we are referring to perennial sea ice only.

Page 19, Section 4.3. This is an interesting idea, but it misses some fundamental characteristics of snow, the most significant being light penetration in snow vs. blue ice, the existence of a saline, damp layer at the base of the snowpack and icy layers and lenses within the snowpack. The Brandt and Warren (1993) study shows that visible wavelengths are not absorbed at depth in a snowpack, but are scattered back to the surface. Near-infrared wavelengths only get absorbed in the top few millimeters of the snowpack. The study then describes optimal conditions where sub-surface melt could be important. These are low-albedo ice like blue ice and low-density snow, like depth hoar. Both conditions are not typical of snow on Antarctic sea ice. The description on page 20, lines 11-16 must be a misinterpretation of the Brant and Warren analysis and needs revising.

We disagree with the reviewer and are confident that we do not misinterpret the Brant and Warren study. We are clear about the importance of extinction coefficients, and the debate about how strong the sub-surface temperature maximum can be. However, we have added a few words to actually suggest that properties of metamorphic snow on Antarctic sea ice can be comparable to the properties of blue ice (which is essentially clear ice with few air bubbles), in as they have large grains lowering the snow's specific surface area (SSA), and ice layers which are essentially clear ice with few air bubbles.

Changed text: Similar internal melting is observed in, e.g., blue ice regions on the Antarctic ice sheet (e.g. Brandt and Warren, 1993;Cheng et al., 2003;Liston and Winther, 2005). While the magnitude of the difference between the snow surface temperature and the sub-surface temperature is debated (Brandt and Warren, 1993), the subsurface temperature maximum depends on the snow's extinction coefficient and is larger with denser snow, with larger grains lowering the specific surface area (SSA), and in the presence of ice layers, which are typical for perennial Antarctic sea ice (Nicolaus et al., 2009).

Related, several studies have since shown technical issues with radiative heating of sensors, such as in Cheng et al. 2003, making observed sub-surface temperature increases somewhat dubious.

We acknowledge the fact that temperature measurements in warm snow in a radiative environment can be challenging. However, we do not rely on measurements for developing our conceptional, qualitative model, and Cheng's modeling and other considerations suggest that the occurrence of subsurface temperature maxima in the snow are quite possible. This is also demonstrated by the subsurface melting on blue ice, even though albedo and extinction coefficients may not be directly comparable with sea ice as discussed above. As witnessed by the authors on Novo airbase, water pockets under a surface layer of ice are quite common during summer, resulting exactly from the vertical profiles of radiation absorption and emission discussed here.

Secondly, there is a wealth of papers that show the widespread occurrence of icy layers and a damp, saline basal layer in snow on Antarctic sea ice, in contrast to an assumption of dry snow as on page 20, line 33. This damp layer, as well as internal icy layers within the snowpack, greatly modify the electromagnetic signature of snow, its temperature gradient and snow metamorphism. I encourage the authors to give these aspects consideration and incorporate them into the proposed hypothesis. If the hypothesis in Section 4.3 conflicts with typical characteristics of Antarctic snow, then explicitly state that it does and describe specifically which environmental conditions the hypothesis is limited to.

We feel that throughout the text and in our replies above we have argued sufficiently and have changed the text sufficiently to acknowledge the presence of damp, saline snow in winter, but have also convincingly explained that saline snow does not play a role in summer. On the contrary, ice layers, which are mentioned by the reviewer along

with saline snow although they usually do not form concurrently in the same seasons, do contribute to the observed backscatter increases and most likely also play an important role in the development of a subsurface temperature maximum.

Although simple, the schematic in Garrity (1992) is informative and may help with this.
We are well aware of the work of Garrity and have worked with her in the field. However, we find that work little useful for our study as it describes only the short time of snow wetting during short melt events, and does not consider the temporal changes throughout the melting period and refreezing. The reported floes were in the marginal ice zone. We suspect that most of that ice did not survive the summer.

For figure 10, it would be helpful to overlay snow grain symbols so that readers can have a better idea of which melt-induced characteristics you're referring to in the snowpack for each stage. The WMO and Colbeck (1991) would be useful references for this.
Unfortunately, there is not one symbol for metamorphic snow but they are distinguished for different grain types which would be misleading at this stage. We therefore decided to keep the given symbols but added horizontal arrows between the day- and nighttime temperature profile in order to make the thaw-freeze cycle clearer.

Page 21, lines 24-27. Could you provide some references to support these statements? Also, what is meant by "the most potential surface changes?"
We have rephrased the sentence and also added a reference for the role of the ocean in contributing to Antarctic sea ice extent variability: However, we believe that they will not, based on the facts that previous changes of ice extent during our study period did not strongly affect melt onset dates as mentioned above, that those recent changes probably have a strong contribution from oceanic processes (Turner et al., 2015), and that most of the potential surface changes related to ice extent fluctuations may occur in the seasonal ice zone or closer to the marginal ice zone where our algorithm is not applicable.

Page 21-22, lines 30-32/lines 1-2. How do we know this is true? Are there references to support these statements?
We don't understand this comment. The paragraph is a summary of our findings and conceptual model and has been supported by the discussion throughout the text.

Page 22, line 3. It's stated that the results obtained in this study demonstrate the potential to observe snow processes at different depths from space. This is not true. However, the study does hypothesize that this could be possible, which is different from a demonstration. Please correct this for clarity.
Rephrased: Based on the results obtained here, we suggest that there is a potential to observe snow processes at different depths from space, opening new avenues for studies of energy and mass budgets of snow on sea ice in the Southern Ocean.

---

## Author Comment (AC2) · 2 Apr 2019

General Comments:

The authors examined the onset of snow melt over Antarctic sea ice using data sets from scatterometers (ERS-1/2, QSCAT and ASCAT) and passive microwave radiometers. Between 1992 and 2015, they found insignificant changes in onset dates which they claim be consistent with the small trends in Antarctic sea ice extent. Also, they used the differential lag in onset timing between the observing instrument to develop a conceptual model for inferring the evolution of the depth/temperature-dependent snow processes during the onset period, and conclude that multi-wavelength instruments may be able to provide information on the behavior of the snow column on Antarctic sea ice during early melt.

Throughout the paper, the authors contrasted the behavior of the Arctic and Antarctic – I find those discussions to be interesting and useful.
We highly appreciate the great work that the reviewer put into revising our manuscript and we realized that he/she is really familiar with the topic of our manuscript. Therefore, we thank the reviewer, as the constructive comments, improved significantly the quality of our manuscript.

The only comment is that the authors inferred from only a few samples (12) the general behavior of the onset-dates of circumpolar ice cover and their relationship to the observed trend in ice extent. This is less credible without more justification as to why a non-uniform sampling of the ice cover (Fig. 1) is sufficient for this analysis.
We had actually conducted our analysis for individual pixels and groups of 3x3 pixels. We agree that a larger region of 3x3 pixels will provide more representative results for the respective areas. Therefore, we are now reporting the results of the analysis of the 3x3 pixel regions. However, this has no effect on the presented results in the manuscript but the given dates shifted slightly by 1-2 days back or forward. However, we refrain from applying the analysis to the entire Antarctic sea-ice area, as there is, so far, no reliable data set on the respective ice types in the ice-covered Southern Ocean.

I would like the comments below addressed prior to publication.

Detailed Comments:
Page:Line number

1:28 While it is true that the circumpolar ice extent has changed insignificantly over the period of study, the trends are significant in the different sectors (e.g., Ross Sea sector). I think that fact should be noted and there are implications as far as the discussions in the remainder of the text regarding expected trends in the onset dates and ice extent.
We agree that it is worth mentioning the regional differences in the temporal evolution of the sea-ice extent over the past decades. We therefore added this fact in the manuscript.

3:20 This is in contrast to what is expected in the Arctic, where backscatter from perennial ice is expected to decrease during the summer. Perhaps another point to note.
This point is already noted in the beginning of page 3 and is therefore not noted again.

6:19 Need clarification: Is it the daily product that was used or the twice daily product? On p.6, the text indicated only the daily product is used.
The word "daily" was wrong and misleading at that point. As mentioned in the following line, QSCAT is given as a twice-a-day product, which is also used here for the analysis of diurnal variations in the snowpack.

7:10 Please specify which ice concentration product is used here.
We used the Bootstrap sea-ice concentration data product. We added this information.

7:15 Perhaps it's good to point out how the samples were 'carefully' chosen to reflect/represent the large-scale behavior/trends of the Southern Ocean ice cover.
The chose locations were the same as those from Haas (2001) in order to ensure both a proper comparison between both studies and a reasonable continuation of the given time series. To point that out, we added the respective reference.

9:1 I don't think 'extensive' is appropriate here – let the reader decide.
We agree and deleted that word.

Figures 3&4. Shouldn't the seasonal ice in sample D disappear during the summer?
Yes, it does disappear. That's why the time period between end of December and end of March is shaded gray in Figure 4b.

Figure 5. Are the backscatter for both C-band and Ku-band merged here? If so, please indicate so because one would expect differences between the two wavelengths.
Yes, we added that.

18:8 Large oceanic heat flux (I imagine relative to the Arctic) – a reference is appropriate here.
Yes, we added a respective reference: Martinson, D. G., and R. A. Iannuzzi (1998), *Antarctic Ocean-ice interaction: Implications from ocean bulk property distributions in the Weddell Gyre*, Wiley Online Library, doi:10.1029/AR074p0243.

18:13 meaningful for? For indicating the ice extent?
We referred that statement to the overall upcoming analysis and discussion (as e.g. sea-ice season duration). To make this clearer, we added this to the given sentence.

18:24 I think this is what you are referring to above, i.e., 18:13.
See previous comment.

Section 4.2 the suggestion here that snow processes may be secondary in explaining the ice extent is important – perhaps worth noting in the abstract. Your work points to that and it is geophysically important, but saying that it is 'NOT' important may be a bit too strong without more discussion and supporting evidence.
We agree and weakened the given statement in the respective context. However, we do not see the need to add this to the abstract as the main focus should be kept on the seasonal snow processes.

20:14 This conceptual model depends on the initial changes to occur in the subsurface prior to that on the surface such that permittivity changes in the interior, while in the pendular regime, leads the change at the surface. I think this is perhaps too dependent on the temperature argument. Is the temperature profile entirely necessary?
Temperature is tbe key physical indicator for snow metamorphism and variations in liquid water content, and therefore we believe that we can not argue with heavily relying on the snow temperature profile and its diurnal and seasonal variations.

---

## Author Response (AR2)

**Anonymous Referee #1, 2nd review report**

I thank the authors for their time spent in carefully addressing the concerns raised in the first review. The revised manuscript is much improved and quite informative. Please find comments below that I hope will be found useful.

Page 1, line 24. I suggest changing "affects" to "may affect" given that it is a conceptual model. We agree and changed that.

Page 1, line 26. I suggest changing "of thick Antarctic snow" to "of thick snow on perennial Antarctic sea ice" since these are the limiting criteria for applying the conceptual model, which is important to convey. We agree and changed that.

Page 2, line 16. Please consider adding "positive" before albedo-feedback here. We added this.

Page 4, lines 11-13. It would be important to note that snow salinity is not negligible during the spring period. Observations by Eicken et al., 1994 and Worby and Massom (1995) show larger ranges in snow salinity during spring. Also, Massom et al., 1997 observed values of 8 psu in summer. Is this negligible for passive microwave, or just low? We adjusted that sentence towards the suggestion of a low salinity which is negligible for satellite microwave observations. However, with due respect, the Eicken et al., 1994 and Massom et al., 1997 studies (to which one of us contributed; Massom et al. (1997) has winter in the title) were carried out in winter and do therefore not show saline snow in spring or summer. Massom et al., 2001 show that snow salinities in summer are mostly 0 ppt. We continue to write in the text that brine percolates out of the snow once snow temperatures rise.

Page 3, line 25. To clarify my earlier concern, please include a statement in the manuscript that these locations may not fully represent multiyear or seasonal sea ice regimes since there may be inter-annual variability in the coverage of ice types at any of these locations. Before, I presumed you had used an ice type product to distinguish multiyear from seasonal ice regimes at these locations.

We added this in the masnucript.

Page 5, line 5. Do you mean 8 rather than 9 nearest neighbors? What is the 9th neighbor? It's an average of in total 9 points, as there is the grid cell with the respective location and then 8 adjacent grid cells.

Page 7, lines 30-33. My concern here is that flooding is not limited to areas with 30% sea ice concentration, and this should be noted in the text. It would be reasonable to state that the effects of flooded ice on the passive microwave signature cannot be fully accounted for in the analysis given the limitations of space-borne observations. We have clarified once again that we limit our analysis to perennial ice, where flooding is very rare as expressed earlier on and which is supported by several references: "...and therefore flooding and saline snow are uncommon on perennial ice in summer (e.g. Massom et al., 2001; Haas et al., 2001; Nicolaus et al., 2009),..."

Page 8, figure 4. It would be helpful to show the corresponding sea ice concentrations as additional figures in supplementary information due to their large influence on backscatter, rather than only showing when concentrations are less than 70%.

We disagree that ice concentration has generally large influence on backscatter as stated by the reviewer. However, low ice concentration and large open water regions can lead to large temporal variations of backscatter, e.g. due to windroughening of the water surface. By limiting our analysis to ice concentrations above 70% we therefore reduce one source of uncertainty as expressed above (Drinkwater and Liu reference). We therefore refrain from showing additional ice concentration time series as we feel they do not add to the conclusions of our study.

Page 9, lines 9-10. "...or the appearance of liquid water in the pendular regime in the middle or lower snowpack." My concern here is that this doesn't appear to be an observationally-based statement, but rather a speculation that liquid water occurs in the middle or lower snowpack first. How do we know that liquid water first occurs in these particular levels of the snowpack? Please state that this is speculation or give references to support this interpretation. We have removed reference to particular layers in the snow column and clarified that the pre-melt backscatter rises can be a result of the appearance of small amounts of liquid water (traces) in the snow, in contrast to the much larger melting and funicular regime that occur at melt onset.

Page 10, line 13. I'm sorry, but it's still not clear what you mean by regionally adaptive here. The paragraph describes an approach for each specific location, but how is this translated to a regional approach? We agree that "regional adaptive" is misleading. We therefore deleted these words as the algorithm is anyway explained after the colon.

Page 12, line 14. It would be helpful to state here that the values differ that those in Haas (2001) due to the different

number of analysed pixels between studies, in case a person would like to reproduce the analyses.

We believe our new results of a backscatter amplitude of 5.8 dB are in good agreement with Haas (2001; 5.6 dB) and are not much affected the different analyzed pixel numbers. However, our studies cover two slightly different periods (1991 to 2001 vs. 1991 to 1999), and we have expressed this in the text.

Page 19, Section 4.3. There are errors that need to be addressed in this section, which are listed below. My concern is that parts of the argument are not supported by the literature and that previous works are referred to inappropriately. While the authors present an interesting concept, more transparency is needed on the knowledge gaps and uncertainties on this topic.

We have now added a reference to Liston et al. (1999) who provide more evidence for subsurface melt and also refer to the Brandt and Warren study.

- The Brandt and Warren, 1993 study did not observe sub-surface maximum temperatures or melt in snow. Other works refer to this analysis as a theoretical modeling study. Rather, the study demonstrates how, once the radiative transfer model's spectral resolution is improved, the sub-surface maximum temperature does not materialize in a snowpack. Rather, they show this phenomenon to be limited in blue ice only. It is erroneous to refer to this work as an observational study of sub-surface temperature maximum and melt.

We have now taken the weight off the Brandt and Warren study and rely more on the observations and modeling of Liston et al. (1999) and Liston and Winther (2005). The combination of these studies also allows us to point out the importance of snow properties and ambient environmental conditions for the occurrence of sub-surface melt. We use this to support our argument that snow properties on perennial Antarctic sea ice are supportive of sub-surface melt.

- Page 20 lines 24-29. The argument that blue ice and melting snow are similar in their inherent optical properties is not supported by the literature and should be noted. The optical properties of melting snow drastically differ from those of blue ice. Blue glacial ice is the most transparent natural medium on Earth. Snow is one of the most reflective mediums on Earth. There is a wealth of literature on this topic, with one of the clearest examples shown in figure 16 of the following paper, which compares the extinction coefficients of blue sea ice (which is relatively less transparent than glacial blue ice) to melting snow: https://apps.dtic.mil/dtic/tr/fulltext/u2/a310586.pdf

Thank you for this interesting reference from one of the most renown sea ice optics experts. We have never claimed that the optical properties of melting snow and blue ice are the same. However, we have referred to blue ice because the phenomenon of subsurface melt is more prominent and widespread there. Next we have expressed that the concepts are applicable to sea ice too, while the magnitudes of melting are much less than in blue ice. We still believe that our hypothesis and reasoning are applicable, and we hope to have convinced the reviewer by adding the Liston references which leave no doubt that subsurface melt in snow is possible.

- The Cheng et al., 2003 states that their the model results were difficult to verify experimentally due to the problems encountered in measuring the temperature profile using temperature sensors embedded in the snow and ice. Too much weight is placed on this study without considering the caveats of radiative heating of sensors, which Cheng describes. We still disagree that we put too much weight on the Cheng et al. (2003) study. We refer to it because of its computation/modeling results, and do not rely at all on temperature observations. As expressed in the previous reply, we are well aware of the difficulty of in-situ snow temperature measurements, and do not rely on that study in the sense that their observations show subsurface melt. It is their modeling that shows it and that is what we clearly refer to. For the same reason we do not put forward our own temperature measurements during the ISPOL project (Haas et al., 2008). However, the satellite measurements we refer to in our study do not destruct the snow or are affected by radiation in any way, and are therefore potentially the best means to detect snow processes and near melting conditions.

Page 20, line 19. Massom et al., 1997 found summertime values of 8 psu, which seems relatively high for snow. I suggest either changing negligible to "low" or explaining what a negligible amount of snow salinity is in relation to, e.g. negligible for detecting a change in the passive microwave signature.

The Massom et al., 1997 study to which one of us contributed was carried out in winter and therefore the reviewer's argument is not suitable. We therefore prefer to keep the statement.

Page 23, lines 10-22. Somewhere within this section, it would be important to state that this conceptual model applies to thick snow on perennial sea ice in the Antarctic. That's a helpful finding to relay to the community. We agree and added this information in the beginning of the explanation of the conceptional model.

[revised manuscript text omitted]

---

## Author Response (AR3)

**Editor Decision: Publish subject to minor revisions (review by editor) (23 Jun 2019) by Martin Schneebeli**

Dear Dr Arndt, dear Steffi
Thank you for the revisions, which are complete, so your manuscript is close to becoming ready for publication. However, I miss a data statement in your manuscript see
https://www.the-cryosphere.net/about/data_policy.html
Please update your manuscript accordingly.
Best regards
Martin Schneebeli
Editor TC

Dear Mr. Schneebeli, dear Martin, thanks a lot for the positive evaluation of our manuscript. We also thank you for the hint of the missing data link. We just uploaded the data to PANGAEA and added this to the acknowledgements. As soon as the complete link is set up, we will add this to the manuscript within the typesetting procedure.

Thanks again for the positive evaluation and best wishes,
Stefanie and Christian

[revised manuscript text omitted]